



# Towards real-time optimal control of wind farms using large-eddy simulations

Nick Janssens and Johan Meyers

KU Leuven, Department of Mechanical Engineering, Celestijnenlaan 300A, B3001 Leuven, Belgium

**Correspondence:** Nick Janssens (nick.janssens1@kuleuven.be)

**Abstract.** Large-eddy simulations (LES) are commonly considered too slow to serve as a practical wind farm control model. Using coarser grid resolutions, this study examines the feasibility of LES for real-time, receding-horizon control to optimize the overall energy extraction in wind farms. By varying the receding-horizon parameters (i.e. the optimization horizon and control update time) and spatio-temporal resolution of the LES control models, we investigate the trade-off between computational speed and controller performance. The methodology is validated on the TotalControl Reference Power Plant using a fine-grid LES model as a wind farm emulator. Analysis of the resulting power gains reveals that the performance of the controllers is primarily determined by the receding-horizon parameters, whereas the grid resolution has minor impact on the overall power extraction. By leveraging these insights, we achieve near-parity between our LES-based controller and real-time computational speed, while still maintaining competitive power gains up to 40 %.

## 1 Introduction

Turbine-wake interactions can significantly impact the efficiency of energy extraction when many turbines are clustered together in large-scale wind farms. Standard control strategies do not anticipate for the wake interactions, but maximize performance at turbine level, resulting in significant power deficits and increased loading in downstream regions. In the last decade, much research has been done into dynamic receding-horizon optimal control strategies to mitigate these effects, both through axial induction control and wake redirection (see Meyers et al. (2022) for a recent review). More recently, Goit and Meyers (2015), Goit et al. (2016) and Munters and Meyers (2017, 2018b) have developed an optimal control framework for wind farm power maximization based on high-fidelity large-eddy simulations (LES) of the wind farm boundary layer. In their latest work combining overinduction and wake redirection control, Munters and Meyers (2018b) report energy gains of up to 34 % for an aligned $4 \times 4$ wind farm. Despite its accuracy, LES is usually deemed impractical for real-time applications due to its high computational cost. In that sense, the aforementioned optimal control studies were intended as benchmarking studies, aimed at exploring the potential of LES for power optimization in wind farms.

To achieve real-time optimal control, one option is to resort to models that are less computationally intensive (compared to 3D LES). For instance, in some dynamic flow models, the vertical dimension of the flow is either disregarded or approximated to cope with the computational cost of 3D wake dynamics (see e.g., Soleimanzadeh et al., 2014; Rott et al., 2017; Boersma et al., 2018). This results in a 2D LES-like model suitable for online wind farm control. Instead of LES, more simplified formulations





of the governing equations, such as the 2D dynamic wake meandering model (Jonkman et al., 2017) or the Reynolds-Averaged Navier–Stokes equations (Iungo et al., 2015), can be employed to accelerate the computations. In Shapiro et al. (2017), an even simpler 1D wake model is proposed for closed-loop receding-horizon control. However, these expeditious engineering models potentially lack the necessary physical intricacies inherent to 3D LES, and may not capture the actual turbulent wake
dynamics.

The present manuscript is a first investigation on the feasibility of using LES as a real-time plant model for receding-horizon wind farm control. To overcome the challenge of computational speed, this study aims to leverage the insights from the earlier work of Bauweraerts and Meyers (2019). In the context of turbulent flow forecasting in the atmospheric boundary layer, they demonstrated that prediction errors only slowly increase when coarsening the grid. By resorting to coarser grid formulations
and incorporating an efficient spatial parallelization, they were able to reduce LES walltimes up to a factor of 300 compared to simulated time. We envisage a similar approach, but focusing on LES-based receding-horizon control. By varying the spatio-temporal grid resolution of the LES plant model, we investigate the trade-off between computational speed and performance of the controller. In view of the latter, we also study the influence of the parameters of the receding-horizon framework, i.e. the optimization horizon, the control update time and number of optimization iterations. To take into account the computational
times for the optimization and allow for a practical, real-time control action, the framework from Munters and Meyers is applied in a time-decoupled fashion where the control signals are computed ahead of time (based on a prediction of the future state, see e.g. Grüne and Pannek, 2017). The proposed methodology is demonstrated on the TotalControl reference power plant (TCRWP) (see e.g. Andersen et al., 2018), combining yaw and induction control strategies.

In the context of wind farm modeling, the coarse LES models envisioned in this work may not fully capture all the relevant
dynamics in the turbulent wakes. In general, finer grids are required to accurately represent secondary flow features such as (the breakdown of) tip vortices and helical structures. For example, in the dynamic induction control (DIC) strategy proposed by Munters and Meyers (2018a), one of the main mechanism to enhance wake mixing is the periodic shedding of vortex rings from front-row turbines synchronized to the turbulent inflow. These vortex rings cannot be accurately represented on a coarse resolution. More novel approaches, such as the helix-approach, also require finer grids to represent the helical structures in
the wakes, especially near the turbine rotor (Frederik et al., 2020). However, for other phenomena that are mostly triggered by large-scale motions in the flow, coarser grids may suffice. Examples of the latter include the overall deflection and gross behavior of the wakes, and potentially also wake meandering triggered by the time-varying inflow conditions or incited by dynamically yawing the turbines (Meyers et al., 2022).

The paper is organized as follows. In Sect. 2, we first summarize some important aspect of the LES modeling and introduce
the time-decoupled receding-horizon framework, and how this framework is adapted to incorporate the coarsening strategy from Bauweraerts and Meyers (2019). Next, the TCRWP test case and simulation setup are discussed in Sect. 3. In Sect. 4, we present the results and discuss the influence of the grid resolution and receding-horizon parameters on the performance and computational time of the LES-based controllers. Section 5 then leverages these insights to design a competitive controller (in terms of power gains) as close to real-time as possible. Section 6 concludes the paper and summarizes the main contributions.





## 2 Methodology


We first discuss the time-decoupled receding-horizon optimal control framework in Sect. 2.1. Next, we describe the wind farm optimization problem in Sect. 2.2 and highlight some aspects of turbine modeling in Sect. 2.3. The optimization method and gradient computation are discussed in Sect. 2.4. Finally, the grid coarsening strategy is elaborated in Sect. 2.5.

### 2.1 Receding-horizon Optimal Control

Previous LES-based wind farm control studies (see e.g. Goit and Meyers, 2015; Goit et al., 2016; Munters and Meyers, 2017, 2018b) have adopted a simplified model-predictive control (MPC) framework where the optimization is performed on an accurate control model assuming perfect knowledge of the system state, see Fig. 1(**a**). This control loop was applied in a receding-horizon fashion (see Fig. 1(**b**)), where time is divided into overlapping windows of length $T_A$. In each consecutive window, the controls $\varphi$ were optimized over a prediction horizon $T$ through a series of fine-grid forward and adjoint LES

simulations, and then applied to the wind farm over a control update time $T_A$ (with $T_A < T$). These controllers, however, are not realizable in practice, because full state information is not available and the fine-grid LES control model is unfeasible due to its excessive computational cost. Moreover, in their framework, the computational time to solve the optimization problems is ignored, since controls are first computed and then applied to the same time interval, which is not possible in real-time.

To account for computational time and allow real-time control, we propose a time-decoupled approach where controls are
computed with a predefined offset corresponding to the control update time $T_A$ from previous studies. The time-decoupled MPC loop, including state estimation, is shown in Fig. 2(**a**) (Grüne and Pannek, 2017). For $t \in [t^k, t^k + T_A]$, the estimator yields an estimate $\hat{\boldsymbol{q}}(t^k)$ of the instantaneous flow at $t^k = kT_A$. The estimate is computed using LiDAR or SCADA measurements $h(\boldsymbol{q}(t))$ that were collected over the past estimation window $[t^k - T_{SE}, t^k]$ and stored in the estimation buffer (with $h(\cdot)$ the measurement function), assuming moving horizon estimation with horizon $T_{SE}$ (note that the MPC loop may differ on the estimation

side when using other approaches such as e.g. a Kalman filter). Next, the predictor uses a flow model to propagate the estimate, yielding a prediction $\hat{\boldsymbol{q}}(t^k + T_A)$ of the future wind farm state. This allows the optimizer to compute the optimal controls ahead of time, resulting in $\varphi_{k+1}(t)$ for $t \in [0, T]$ (with $T$ the optimization horizon). This set of controls is then stored in the actuator buffer. In the next window ($t \in [t^k + T_A, t^k + 2T_A]$), the subset $\phi(t) = \varphi_{k+1}(t - (k+1)T_A)$ of controls corresponding to that window is released by the buffer and applied to the farm. In a real-time setting, the update time $T_A$ should be long enough to

compensate for the computational time of estimation, prediction and optimization. Similar schemes have been proposed in e.g. Chen et al. (2000), Findeisen and Allgöwer (2003) and Su et al. (2013) for low-dimensional generic demonstration problems.

The time-decoupled control loop can be applied in a receding-horizon framework. The sequence of real-time computations versus the corresponding receding-horizon computations (in simulated time) is outlined in Fig. 3. As described above, the state estimate is typically generated based on past flow measurements over some estimation horizon $T_{SE}$, whereas the optimization

is performed over a time horizon $T$ with offset $T_A$.

To reduce the computational time, similar as in Bauweraerts and Meyers (2019), the prediction and optimization are performed using a coarse grid wind farm model. The actual wind farm in Fig. 2(**a**) is represented by an emulator in the form of





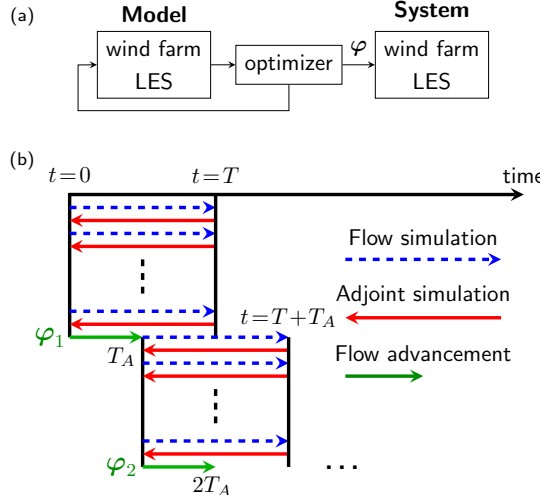

**Figure 1.** MPC approach from earlier studies (Goit and Meyers, 2015; Goit et al., 2016; Munters and Meyers, 2017, 2018b). (**a**) Control loop. (**b**) Receding-horizon approach. In every window $i$, the optimization stage is represented by a series of forward and adjoint simulations, resulting in controls $\varphi_i$ that are applied to the farm over a control update time $T_A$. Figures adapted from Munters and Meyers (2018b).

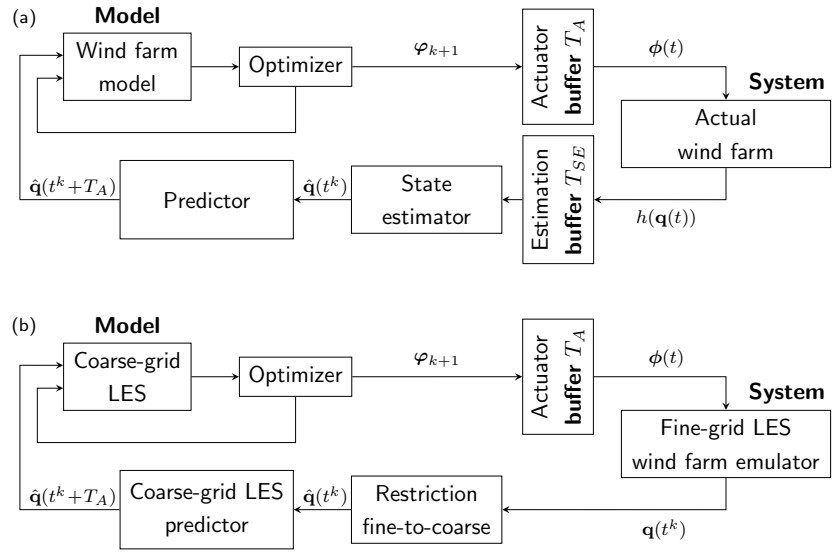

**Figure 2.** (**a**) Time-decoupled MPC assuming moving horizon estimation. In the window $t \in [t^k, t^k + T_A]$, a state estimate $\hat{q}(t^k)$ is computed based on past measurements $h(q(t))$ stored in the estimation buffer; controls are computed with offset $T_A$ based on a prediction $\hat{q}(t^k + T_A)$ of the future state, and buffered until they are applied in the next window $t \in [t^k + T_A, t^k + 2T_A]$. (**b**) Time decoupled MPC approach considered in this work, assuming perfect state information. Controls and prediction are computed using coarse-grid LES, fine-grid LES serves as an emulator for the actual farm.





fine-grid LES, to which the control action $\phi$ is applied. In the present study, we assume perfect state knowledge and hence omit
the state estimator. Instead, we just introduce a restriction operator to filter the exact wind farm state onto the coarse prediction
and optimization grids. Under this simplification, the exact state (but restricted to the coarse grid) is then propagated forward
in time by $T_A$ in the predictor to generate the prediction $\hat{\boldsymbol{q}}(t^k + T_A)$ of the future wind farm state. The restriction operator, used
to map system feedback from the fine LES grid to the coarser prediction and optimization grid, is discussed in more detail in
Sect. 2.5. The time-decoupled control loop, omitting state estimation, is shown in Fig. 2(**b**). Remark that this approach intro-
duces two sources of model mismatch between the coarse models and the fine-grid emulator. Firstly, a restriction error arises
from filtering the LES data from the emulator to the coarser resolution of the predictor and optimizer. Secondly, the predictor
introduces a prediction error that is inherent to time-decoupled MPC and depends on the update time $T_A$.

## 2.2 Wind Farm Optimization Problem

In the receding-horizon approach, in every optimization window, the overall wind farm power is optimized over the optimiza-
tion horizon $T$. The optimization problem is formulated as in Munters and Meyers (2018b):

$$\min_{\substack{\boldsymbol{\varphi}(t),\boldsymbol{q}(t) \\ 0<t\leq T}} \mathcal{J}(\boldsymbol{\varphi},\boldsymbol{q}) = -\int_0^T \sum_{m=1}^{N_t} \frac{1}{2} C'_{P,m} \overline{V}_m^3 A_m dt \tag{1}$$

$$\text{s.t.} \quad \frac{\partial \boldsymbol{u}}{\partial t'} + (\boldsymbol{u}\cdot\nabla)\boldsymbol{u} = -\frac{\nabla p}{\rho} - \nabla\cdot\boldsymbol{\tau}_{\text{sgs}} - \sum_{m=1}^{N_t} \frac{1}{2}\hat{C}'_{T,m}\overline{V}_m^2 \mathcal{R}_m(\boldsymbol{x})\boldsymbol{e}_{\perp,m} \qquad \text{in } \Omega\times(0,T], \tag{2}$$

$$\nabla\cdot\boldsymbol{u} = 0 \qquad \text{in } \Omega\times(0,T], \tag{3}$$

$$\tau\frac{d\hat{C}'_{T,m}}{dt} = C'_{T,m} - \hat{C}'_{T,m} \qquad m=1\dots N_t \text{ in } (0,T], \tag{4}$$

$$\frac{d\theta_m}{dt} = \omega_m \qquad m=1\dots N_t \text{ in } (0,T], \tag{5}$$

$$C'_{T,\text{min}} \leq C'_{T,m} \leq C'_{T,\text{max}} \qquad m=1\dots N_t \text{ in } (0,T], \tag{6}$$

$$\omega_{\text{min}} \leq \omega_m \leq \omega_{\text{max}} \qquad m=1\dots N_t \text{ in } (0,T]. \tag{7}$$

$$\boldsymbol{u}(\boldsymbol{x},0) = \hat{\boldsymbol{u}}_0^{\text{pred}} \qquad \text{in } \Omega, \tag{8}$$

The flow through the wind farm is governed by the LES equations (2)–(3), with $\boldsymbol{u}$ and $p$ the velocity and pressure. Subgrid-
scales are represented by the stress tensor $\boldsymbol{\tau}_{\text{sgs}}$ using a standard Smagorinsky model with constant coefficient $C_s = 0.14$ in-
cluding wall damping. For the discretization, we employ a pseudo-spectral method in the stream- and spanwise direction, and
a fourth-order energy-conservative finite difference scheme in the vertical direction. For the time stepping, we use an explicit
fourth-order Runge–Kutta scheme. Here, all four Runge–Kutta stages are stored on disc (as opposed to the aforementioned
LES studies that only store the first stage). This results in a more accurate representation of the gradients using the adjoint
method (see Sect. 2.4). In the vertical direction, a high Reynolds number wall stress boundary condition and symmetry bound-
ary condition are imposed on the bottom and top surface of the domain respectively. All simulations are performed using the
in-house simulation code SP-Wind (for more information, see e.g. Goit and Meyers, 2015; Munters and Meyers, 2017, 2018b).



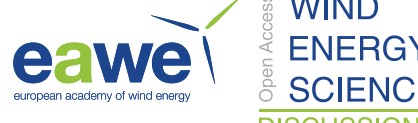

**Figure 3.** Sequence of real-time computations (estimation, prediction and optimization) versus corresponding receding-horizon computations in simulated time, corresponding to the control loop from 2(**a**).




Each turbine $m$ $(=1\ldots N_t)$ is controlled by a time-dependent thrust coefficient setpoint $C'_{T,m}$ and yaw rate $\omega_m$, both subject to box constraints (6)–(7). Put together, they constitute the vector of optimization variables: $\boldsymbol{\varphi}(t) = [C'_{T,1}(t),\ldots,C'_{T,N_t}(t),\omega_1(t),\ldots,\omega_{N_t}(t)] = [\boldsymbol{C}'_T(t),\boldsymbol{\omega}(t)]$ for $0 < t \le T$. To model the turbine response time, an exponential filter (4) with time constant $\tau$ is applied to the thrust coefficient setpoints, resulting in the time-filtered disc-based thrust coefficients $\hat{\boldsymbol{C}}'_T(t)$ (Munters and Meyers, 2017). The yaw rates control the turbine yaw angles $\boldsymbol{\theta}(t)$ through the yaw equation (5). All together, this results in the vector of state variables $\boldsymbol{q} = [\boldsymbol{u}(\boldsymbol{x},t), p(\boldsymbol{x},t), \hat{\boldsymbol{C}}'_T(t), \boldsymbol{\theta}(t)]$. The disc-based thrust coefficients and yaw angles determine the thrust force exerted on the flow and the power extracted by the turbine (see Sect. 2.3). For a more detailed explanation on all the terms and equations, see Goit and Meyers (2015) and Munters and Meyers (2017, 2018b).

Remark that controls are computed ahead of time in line with the time-decoupled MPC loop from Sect. 2.1. In receding-horizon interval $k$ (for $t \in [t^k, t^k + T_A]$, see Fig. 3), the exact state from the fine-grid emulator is first restricted to the coarse grid, resulting in $\hat{\boldsymbol{q}}(t^k)$. Next, the prediction $\hat{\boldsymbol{q}}(t^k + T_A)$ is computed by propagating $\hat{\boldsymbol{q}}(t^k)$ over $T_A$ using the same coarse-grid LES model described by eqs. (2)–(7). This estimate is used as initial condition $\hat{\boldsymbol{u}}_0^{\text{pred}}$ for the optimization, see eq. (8). Also note that the controls $\boldsymbol{\phi}(t)$, that are actually applied to the wind farm emulator in the next time interval (for $t \in [t^k + T_A, t^k + 2T_A]$), only comprise the first part of length $T_A$ from the optimal controls $\boldsymbol{\varphi}_{k+1}(t)$ (with $T_A \le T$).

## 2.3 Wind Turbine Modeling

For the turbine modeling, a standard non-rotating actuator disc model is used. Based on actuator disc theory, the turbines exert a force on the flow: $\boldsymbol{f}_m = -\frac{1}{2}\hat{C}'_{T,m}\overline{V}_m^2 \mathcal{R}_m(\boldsymbol{x})\boldsymbol{e}_{\perp,m}$, where $\mathcal{R}_m$ is a smoothed footprint of the rotor on the LES grid, $\boldsymbol{e}_{\perp,m}$ the unit vector perpendicular to the rotor plane and $\overline{V}_m = \frac{M}{A_m}\int_\Omega \mathcal{R}_m \boldsymbol{u} \cdot \boldsymbol{e}_{\perp,m} d\boldsymbol{x}$ the (corrected) disc-averaged velocity (with $A_m$ the rotor disc area and $M$ a correction factor defined below). The power extracted from turbine $m$ is then given by $P_m = \frac{1}{2}C'_{p,m}\overline{V}_m^3 A_m$, where $C'_{p,m}$ denotes the disc-based power coefficient.

On present-day grid resolutions, power is typically overestimated due to the diffuse smearing of the rotor disc on the simulation grid by the rotor footprint (Martínez-Tossas et al., 2015; Shapiro et al., 2019). To account for this, Munters and Meyers (2017, 2018b) have proposed to set $C'_p = a\hat{C}'_T$, where $a$ is selected based on fitting LES data to 1D momentum theory. While linear scaling is effective on intermediate grid resolutions for unyawed turbines, its performance deteriorates on coarser grids and for yawed turbines. This is illustrated in Fig. 4(a)-(b), where we show the empirical power coefficient $C_p = P/\left(\frac{1}{2}U_\infty^3 A\right)$ for the DTU 10MW turbine for different values of $a$ at yaw angles $\theta = 0°$ and $\theta = 30°$, computed using LES with uniform inflow ($U_\infty = 8 \text{ m s}^{-1}$) on a grid resolution $\Delta x = \Delta y = 1.6\Delta z = 80$ m (the coarsest resolution in this work). We also show the effect of the correction from Shapiro et al. (2019), which is expressed as a correction factor on the disc-averaged velocity $V_m = \frac{1}{A_m}\int_\Omega \mathcal{R}_m \boldsymbol{u} \cdot \boldsymbol{e}_{\perp,m} d\boldsymbol{x}$. In particular, the corrected disc-averaged velocity is given by $\overline{V}_m = M^{\text{sh}}V_m$, with

$$M^{\text{sh}} = \left(1 + \frac{\hat{C}'_{T,m}}{4}\frac{1}{\sqrt{3\pi}}\frac{\Delta}{R}\right)^{-1}, \tag{9}$$

the Shapiro factor, $R$ the rotor radius and $\Delta$ the filter width of Gaussian filtering Kernel. For comparison, the power coefficient (for $a = 1.0$) on the reference resolution $\Delta x = \Delta y = 2\Delta z = 13.33$ m (the resolution of the emulator) is also included, where we also applied the Shapiro factor to better replicate 1D momentum theory.





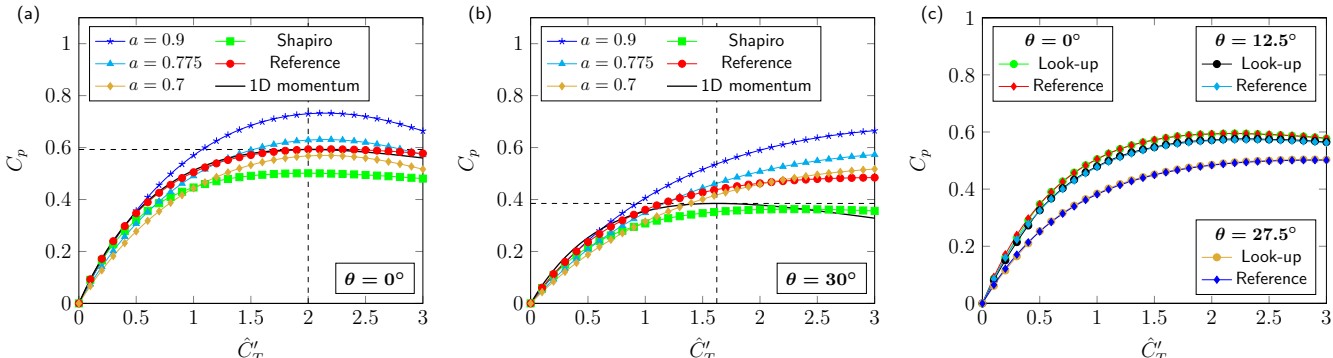

**Figure 4.** Empirical power coefficient versus disc-based thrust coefficient for different correction strategies at different yaw angles $\theta$: (**a**) linear scaling $C_P' = a\hat{C}_T'$, (**b**) factor from Shapiro et al. (2019) in eq. (9), and (**c**) look-up table approach. For every simulation, uniform inflow velocity $U_\infty = 8$ m s$^{-1}$ is prescribed using a fringe region spanning the final 20 % of the domain, with domain size $L_x = 26.92D$, $L_y = 13.46D$, $L_z = 8.41D$ on the coarse grid $\Delta x = \Delta y = 1.6\Delta z = 80$ m. In each plot, we also show the empirical power coefficient (including Shapiro correction) at the reference resolution $\Delta x = \Delta y = 2\Delta z = 13.33$ m.

In Fig. 4(**a**), the correction $a = 0.775$ performs relatively well compared to the reference. However, under yaw misalignment of $30°$ in Fig. 4(**b**), the same factor overestimates power for higher thrust coefficients. The Shapiro correction consistently underestimates power on the coarse grid. In an optimal control setting, any mismatch between model and reference will result in suboptimal thrust coefficients and yaw angles. Therefore, we take $a = 1$ and propose a look-up table approach to account for discrepancies in power prediction on different grid resolutions. In particular, the disc-averaged velocity on the coarse grids

is corrected based on a look-up value $M$ that depends on the thrust coefficient and the yaw angle:

$$\overline{V}_m = M(\hat{C}_{T,m}', \theta_m)V_m. \tag{10}$$

The look-up table is constructed in such a way that, for uniform inflow, the corrected disc-averaged velocity (and consequently the power output) on the coarse grids matches the disc-averaged velocity from the reference for given $(\hat{C}_T', \theta)$. This approach allows to match power calculation on the coarse grids arbitrarily well with a given reference turbine (depending

on the look-up table resolution in terms of $\hat{C}_T'$ and $\theta$). In the current work, we propose a look-up range $\hat{C}_T'^{\text{table}} \times \theta^{\text{table}} = \{0.0, 0.5, \ldots, 2.5, 3.0\} \times \{0°, 5°, \ldots, 35°, 40°\}$, and we use bilinear interpolation on the look-up table to compute the correction for given $\hat{C}_T'$ and $\theta$. The resulting look-up tables are tabulated in App. A. The corrected power coefficient is shown in Fig. 4(**c**).

### 2.4   Optimization Method and Gradient Computation

As in Munters and Meyers (2018b), the optimization problem is solved in a reduced fashion by explicitly substituting state

equations (2)–(5) into the cost function, i.e. by minimizing $\tilde{\mathcal{J}}(\varphi) = \mathcal{J}(\varphi, q(\varphi))$ subject to box constraints (6)–(7). To solve the optimization problem, we use the limited-memory Broyden–Fletcher–Goldfarb–Fanno method with box constraints (L-BFGS-B). In each iteration, this quasi-Newton method constructs a quadratic approximation of the objective function, where



the inverse Hessian is approximated by the BFGS formula. A search direction is then generated based on the minimum of the quadratic model (Nocedal and Wright, 2006). We use a linesearch method to determine a step length in the search direction

that satisfies the strong Wolfe conditions. In every optimization window, this procedure results in a sequence of function and gradient evaluations (cf. Fig. 3). To compensate for the large-scale nature of the problems at hand, we use the limited-memory version of BFGS that only stores a limited number of correction pairs. We use the L-BFGS-B Fortran library from Zhu et al. (1997) and Morales and Nocedal (2011); for more information on the algorithm, the reader is referred to Byrd et al. (1995).

In contrast to previous LES-based wind farm control studies (Goit and Meyers, 2015; Goit et al., 2016; Munters and Meyers,

2017, 2018b) that relied on a continuous adjoint approach to compute gradients, we follow up on the work in Yilmaz and Meyers (2019) by using a temporally discrete adjoint method. In the continuous adjoint method, the adjoint equations are first derived based on the description of the optimization problem in (1)–(8), and subsequently discretized and solved using LES similar to a forward simulation. Conversely, the discrete adjoint method first discretizes and linearizes the state equations, and then formulates the discrete adjoint of the linearized equations. Consequently, the discrete adjoint method obtains the gradient

of the discretized cost functional, whereas the continuous adjoint method yields a discrete approximation of the gradient of the continuous cost functional. In the limit of infinite grid resolution, both methods are equivalent (Giles and Pierce, 2000).

Below we directly formulate the temporally discrete adjoint Runge–Kutta 4 scheme, derived from a fourth-order Runge–Kutta discretization of the state equations (2)–(5) and cost function. More details on the discretization are provided in App. B1. For the Navier–Stokes equations, the thrust coefficient filter equation and the yaw equation, this results in the following (where

$i$- and $j$-subscripts denote the Runge–Kutta stages, $m$ is the turbine number and $n$ the discrete time instant, i.e. $t^n = n\Delta t$):

$$\boldsymbol{\xi}_i^n/\Delta t = \left(-(\nabla \boldsymbol{u}_i^n)^T + (\boldsymbol{u}_i^n \cdot \nabla)\right)\hat{\boldsymbol{\xi}}_i^n - \nabla \pi_i^n/\rho - \nabla \cdot \boldsymbol{\tau}_{\text{sgs}}^*\left(\boldsymbol{u}_i^n, \hat{\boldsymbol{\xi}}_i^n\right) + \sum_{m=1}^{N_t} \boldsymbol{f}_{m,i}^{*n} \qquad i = 1\dots4, \quad (11)$$

$$\nabla^2 \pi_i^n/\rho = \nabla \cdot \left[\left(-(\nabla \boldsymbol{u}_i^n)^T + (\boldsymbol{u}_i^n \cdot \nabla)\right)\hat{\boldsymbol{\xi}}_i^n - \nabla \cdot \boldsymbol{\tau}_{\text{sgs}}^*\left(\boldsymbol{u}_i^n, \hat{\boldsymbol{\xi}}_i^n\right) + \sum_{m=1}^{N_t} \boldsymbol{f}_{m,i}^{*n}\right] \qquad i = 1\dots4, \quad (12)$$

$$\boldsymbol{\xi}^n = \boldsymbol{\xi}^{n+1} + \sum_{i=1}^{4} \boldsymbol{\xi}_i^n, \qquad (13)$$

$$\sigma_{m,i}^n/\Delta t = \frac{1}{\tau}\left(\hat{\sigma}_{m,i}^n - \frac{1}{2}\overline{V}_{m,i}^{n\,2}(b_i\overline{V}_{m,i}^n - \hat{X}_{m,i}^n)A_m - \frac{1}{2}\hat{C}_{T,m}'^n\overline{V}_{m,i}^n V_{m,i}^n \frac{\partial M(\hat{C}_{T,m}'^n, \theta_m^n)}{\partial \hat{C}_T'}(3b_i\overline{V}_{m,i}^n - 2\hat{X}_{m,i}^n)A_m\right) \qquad i = 1\dots4, \quad (14)$$

$$\sigma_m^n = \sigma_m^{n+1} + \sum_{i=1}^{4} \sigma_{m,i}^n, \qquad (15)$$

$$\eta_{m,i}^n/\Delta t = -\frac{1}{2}\hat{C}_{T,m}'^n\overline{V}_{m,i}^n\left[\int_\Omega \left((3b_i\overline{V}_{m,i}^n - 2\hat{X}_{m,i}^n)M\boldsymbol{u}_i^n - \overline{V}_{m,i}^n\boldsymbol{\xi}\right)\cdot\left(\boldsymbol{e}_{||,m}\mathcal{R}_m + \boldsymbol{e}_{\perp,m}\mathcal{D}_m\right)d\boldsymbol{x}\right]$$

$$-\frac{1}{2}\hat{C}_{T,m}'^n\overline{V}_{m,i}^n V_{m,i}^n \frac{\partial M(\hat{C}_{T,m}'^n, \theta_m^n)}{\partial \theta}(3b_i\overline{V}_{m,i}^n - 2\hat{X}_{m,i}^n)A_m \qquad i = 1\dots4, \quad (16)$$

$$\eta_m^n = \eta_m^{n+1} + \sum_{i=1}^{4} \eta_{m,i}^n, \qquad (17)$$





with $a_{ij}$ and $\beta_i$ the Runge–Kutta coefficients. In these equations, $\boldsymbol{\xi}^n$, $\pi^n$, $\sigma_m^n$, $\eta_m^n$ are the adjoint state variables associated

to $\boldsymbol{u}^n$, $p^n$, $\hat{C}'^n_{T,m}$, $\theta_m^n$, and $\boldsymbol{\xi}_i^n$, $\pi_i^n$, $\sigma_{i,m}^n$, $\eta_{i,m}^n$ for $i=1\ldots4$ are the corresponding adjoint Runge–Kutta stages. Furthermore, $\hat{\boldsymbol{\xi}}_i^n$, $\hat{\sigma}_{i,m}^n$ and $\hat{\eta}_{i,m}^n$ are auxiliary variables (see App. B). As in Munters and Meyers (2018b), $\boldsymbol{f}_m^*$ denotes the adjoint turbine force, $X_m$ is the disc-averaged adjoint velocity, $\boldsymbol{e}_{\parallel,m}$ the rotor-parallel unit vector and $\mathcal{D}_m$ the rotational rotor footprint. The temporally discrete adjoint equations are derived in detail in App. B. For a detailed definition of all terms related to the turbine modeling, see Goit and Meyers (2015) and Munters and Meyers (2018b). However, note that these studies only used the first

forward Runge–Kutta stage, whereas here all four (forward) Runge–Kutta stages are used in the adjoint scheme, resulting in a more accurate method.

Finally, the adjoint variables are used to compute the gradients with respect to the thrust coefficient setpoints and yaw rates:

$$
\nabla_{\varphi_m^n} \tilde{\mathcal{J}}^N = \begin{pmatrix} \frac{\partial \tilde{\mathcal{J}}^N}{\partial C'^n_{T,m}} \\ \frac{\partial \tilde{\mathcal{J}}^N}{\partial w_m^n} \end{pmatrix} = \begin{pmatrix} -\Delta t \sum_{i=1}^4 \left( b_i \sigma_m^{n+1} + \sum_{j=1}^{i-1} a_{ij} \sigma_{m,i}^n \right) \\ -\Delta t \sum_{i=1}^4 \left( b_i \eta_m^{n+1} + \sum_{j=1}^{i-1} a_{ij} \eta_{m,i}^n \right) \end{pmatrix}. \tag{18}
$$

Remark that, as mentioned above, eq. (18) is an exact expression for the gradient of the discretized objective function. The

discrete adjoint approach and gradient computation are validated in App. B5.

### 2.5   Coarse Grid Optimization and Coarsening Strategy

As discussed in Sect. 1, we investigate the influence of the spatio-temporal grid resolution of the LES wind farm control model on the overall power gain and computational speed. To that end, as in Bauweraerts and Meyers (2019), we define three coarse grid resolutions for prediction and optimization, as well as a fine reference grid for the wind farm emulator. For the grid

specifications, the reader is referred to Sect. 3.1 where we discuss the case setup. Below, we discuss the coarsening strategy in view of the time-decoupled MPC loop from Fig. 2(**b**).

#### 2.5.1   Coarse Grid Prediction and Control Methodology

At the start of every window, feedback (i.e. the 3D flow field) from the fine-grid wind farm emulator (the reference) is provided to the coarse predictor through a restriction operator (cf. Fig. 2(**b**)). In view of computational time, we allow different domain

sizes for the LES models in the coarse-grid predictor and optimizer. In particular, for the predictor, we propose an upstream domain length proportianal to the total prediction horizon $T+T_A$, i.e. $L_{x,\text{upstream}} = \alpha(T+T_A)U_\infty$ (where $\alpha \geq 1$ is a safety factor). The predictor uses this domain to propagate the restricted reference field, which in turn yields the initial condition for the optimization. For the optimization, we take $L_{x,\text{upstream}} = \alpha T U_\infty$. Thus, the approach is characterized by cropping and restricting the reference field to the prediction domain, and subsequently another cropping from the prediction to the optimization domain,

as graphically illustrated in Fig. 5. The horizon-dependent upstream domain lengths ensure that inflow never reaches the front-row turbines within the prediction and optimization windows, rendering fringe regions and turbulent inflow generation superfluous in predictor and optimizer. The fringe region from the fine-grid reference simulation is therefore excluded upon restriction which, along with the smaller domain sizes (for a given grid resolution), entails significant computational speed-ups. We note that the lack of proper inflow may influence the optimized controls towards the end of the optimization window.





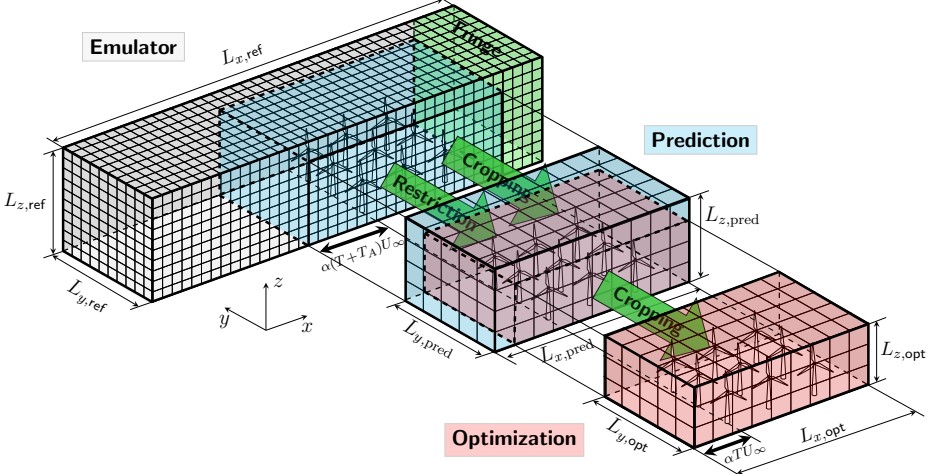

**Figure 5.** Coarsening strategy: the fine-grid reference field is cropped (removing the fringe region and retaining an upstream domain length of $\alpha T U_\infty$) and restricted to the coarser resolution for the optimization. The resulting field is used as initial condition for the optimization.

However, choosing an update time $T_A < T$ should suffice to counter these effects. Also note that, besides the horizon-dependent streamwise cropping, we also allow for a spanwise and vertical cropping to further accelerate prediction and optimization.

### 2.5.2   Restriction from Reference to Optimization Resolution

Given the pseudo-spectral discretization in SP-Wind, the reference flow field from the emulator must be transformed from fourier space into real space before applying the cropping. The cropped reference velocity in real space, $\hat{u}^{\frac{3}{2},\text{ref}}$, is then

restricted to the coarser resolution used in the predictor and optimizer using linear interpolation (superscript 'ref' denotes the reference):

$$\hat{\boldsymbol{u}}^{\frac{3}{2}}(x^{\frac{3}{2}}, y^{\frac{3}{2}}, z) = \sum_{i=1}^{\frac{3}{2}N_x^{\text{ref}}} \sum_{j=1}^{\frac{3}{2}N_y^{\text{ref}}} \sum_{k=1}^{N_z^{\text{ref}}} \hat{\boldsymbol{u}}^{\frac{3}{2},\text{ref}}(x_i^{\frac{3}{2},\text{ref}}, y_j^{\frac{3}{2},\text{ref}}, z_k^{\text{ref}}) \max\left(1 - \left|\frac{z - z_k^{\text{ref}}}{\Delta z^{\text{ref}}}\right|, 0\right) \max\left(1 - \left|\frac{y^{\frac{3}{2}} - y_j^{\frac{3}{2},\text{ref}}}{\frac{2}{3}\Delta y^{\text{ref}}}\right|, 0\right) \max\left(1 - \left|\frac{x^{\frac{3}{2}} - x_i^{\frac{3}{2},\text{ref}}}{\frac{2}{3}\Delta x^{\text{ref}}}\right|, 0\right), \quad (19)$$

where $\hat{u}^{\frac{3}{2}}$ denotes the restricted velocity (in real space) and $\Delta x^{\text{ref}}$, $\Delta y^{\text{ref}}$ and $\Delta z^{\text{ref}}$ the grid resolutions of the reference grid. Note that the restriction takes place in real space, where the grid is a factor $3/2$ finer in the horizontal directions for de-aliasing using Orzag's rule (hence the factors $3/2$ in the equation above).

## 3   Case Setup

This section provides a detailed description of the simulation cases and numerical setup that are used to evaluate the proposed wind farm controller. Throughout this work, we consider the TotalControl Reference Wind Power Plant (TCRWP) (Andersen et al., 2018) consisting of 32 DTU 10MW turbines arranged in an $8 \times 4$ aligned pattern, as illustrated in Fig. 6. Turbines are separated by an intermediate spacing of $5D$ in stream- and spanwise directions, where $D = 178.3$ m is the rotor diameter and

turbines are placed at hub height $z_h = 119$ m (based on the DTU 10MW reference turbine, as reported in Bak et al. (2013)).

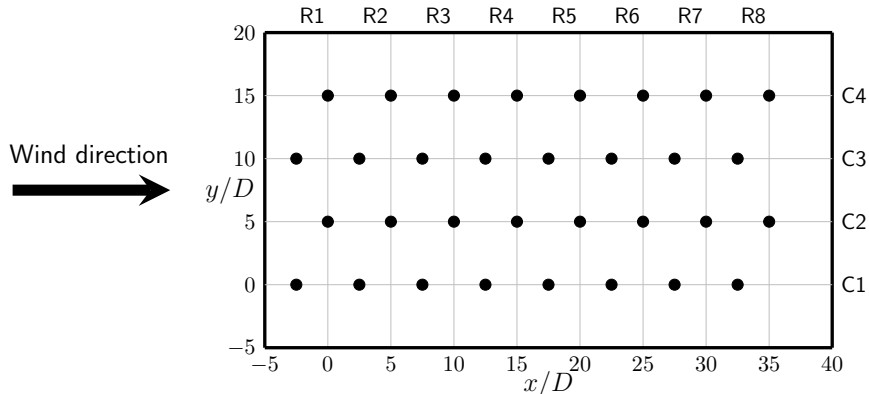

**Figure 6.** Layout of the TotalControl Reference Wind Power Plant. Axes in rotor diameter units, with $D = 178.3$ m. Figure adapted from Andersen et al. (2018).

### 3.1 Simulation Setup

The grid resolutions for the numerical discretization on the three coarseness levels and the reference are summarized in Tab. 1. Since SP-Wind requires a constant integration time step, $T/\Delta t$ should be an integer. Furthermore, we use the same $\Delta t^i$ for all cases on grid level $i$. Therefore, the time steps in Tab. 1 are selected as the largest possible ones meeting these requirements

and adhering to a CFL condition of $0.8$. Note that based on the CFL condition only, the time step on the coarsest grid levels could in principle still be higher hence speeding up the computation. The latter case is investigated in Sect. 5, where we design a controller that is as close to real-time as possible.

For the fine-grid reference simulation in the wind farm emulator, we take the simulation setup from Andersen et al. (2018) and Sood and Meyers (2020), consisting of $1200 \times 1200 \times 225$ grid cells and a simulation domain with dimensions

$L_x \times L_y \times L_z = 16 \times 16 \times 1.5$ km$^3$, where the final $6.25$ % of the domain in the streamwise direction is used as a fringe region to impose the inflow conditions. The spatial extent of the domain suffices to keep blockage effects negligible in the reference simulation. The simulations are performed using a standard offshore roughness length $z_0 = 2 \times 10^{-4}$ m, and the flow is driven by a pressure gradient $\partial_x p_\infty / \rho = 5.2267 \times 10^{-5}$ m s$^{-2}$ resulting in a friction velocity $u_\tau = 0.28$ m s$^{-1}$, which is a typical value in offshore boundary layers.

Prior to the wind farm simulations, a pressure driven precursor simulation with periodic boundary conditions was run on the same (reference) domain to generate turbulent inflow. With the roughness length and pressure gradient specified above, this results in a freestream wind speed roughly equal to $9.4$ m s$^{-1}$ at the turbine hub height. The precursor data (with a detailed overview of the precursor simulation setup) is publicly available in Munters et al. (2019). Using this precursor, the flow is then advanced through the wind farm for a spin-up period of $60$ min to account for startup transients. Remark that here, turbines

are modeled by non-rotating actuator discs, and the turbine locations are shifted backwards in the streamwise direction in comparison to Sood and Meyers (2020). The latter is required to accomodate the entire flow field encompassed in the longest





**Table 1.** Grid resolutions for the different coarseness levels.

| Grid level $i$ | | 0 | 1 | 2 | reference |
|---|---|---|---|---|---|
| Resolution $x$ [m] | $\Delta x^i$ | 80 | 60 | 40 | 13.33 |
| Resolution $y$ [m] | $\Delta y^i$ | 80 | 60 | 40 | 13.33 |
| Resolution $z$ [m] | $\Delta z^i$ | 50 | 37.5 | 25 | 6.67 |
| Resolution $t$ [s] | $\Delta t^i$ | 2.5 | 2.5 | 2.0 | 0.5 |

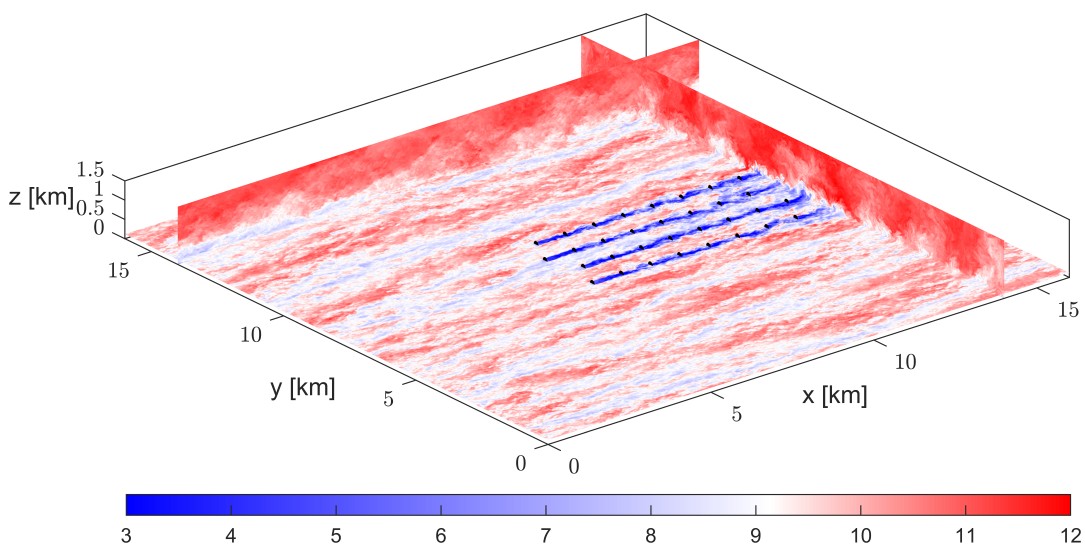

**Figure 7.** Snapshot of the initial condition for optimal control (on the reference resolution). Colors represent the instantaneous velocity magnitude [m s$^{-1}$]. The black dots represent wind turbine locations.

prediction windows (cf. Sect. 2.5). The resulting flow field, depicted in Fig. 7, is used as the starting point for the optimization. Figure 8 shows the initial flow field after restriction to the resolutions of the coarse control models from Tab. 1.

As explained in Sect. 2.5, the upstream domain lengths for the coarse models in the predictor and optimizer are chosen
proportional to the optimization horizon $T$ and control update time $T_A$. In this work, we consider four different horizons (see also Sect. 3.1.1): $T, T_A \in \{50, 150, 250, 350\}$. The corresponding domain sizes and grid specifications for each combination of $T$ and $T_A$ are summarized in Tab. 2 for both predictor and optimizer. In all simulations, a high-Reynolds number wall model is used at the bottom of the domain with roughness length $z_0 = 2 \times 10^{-4}$ m, and the top boundary is treated by a stress-free condition. For the prediction and optimization, we use the same coarse grid resolutions from Tab. 1 and omit the fringe region
(cf. Sect. 2.5).

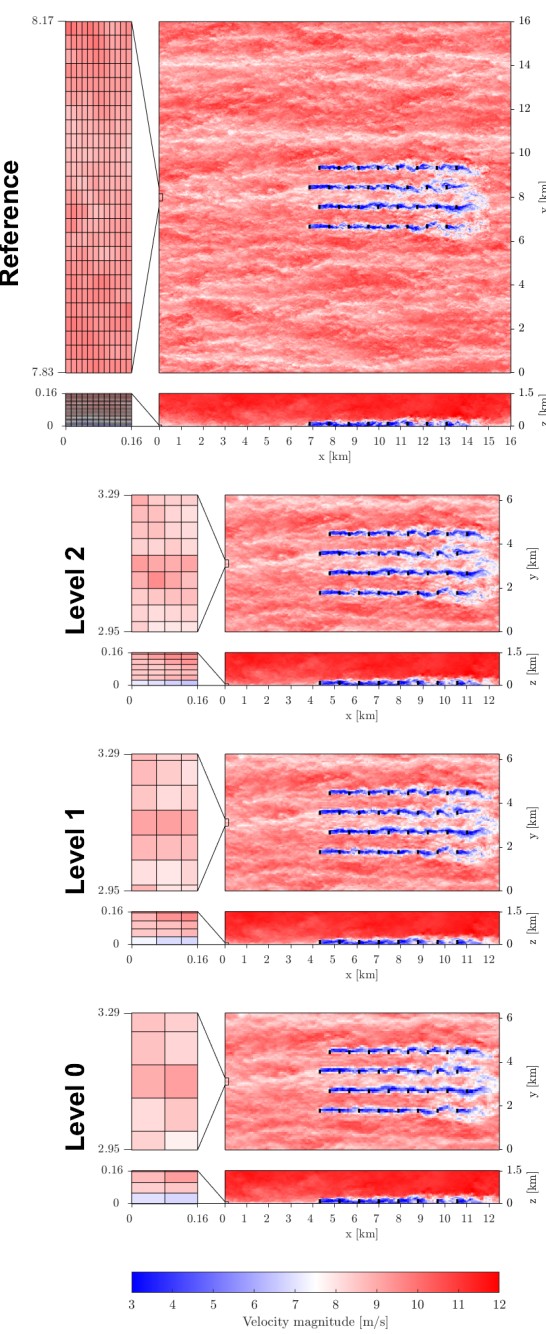

**Figure 8.** Snapshot of the initial condition for optimal control on the reference resolution and coarser optimization resolution from Tab. 1. z-slices are taken at hub height, y-slices at the location of the first column of turbines. Colors represent the instantaneous velocity magnitude [m s$^{-1}$]. The black dots represent wind turbine locations. Grid resolutions are also indicated.



**Table 2.** Grid specifications and domain sizes for the different coarseness levels in function of the optimization horizon $T$ and control update time $T_A$ for optimizer and predictor. Domain sizes in spanwise and vertical directions are equal for all cases: $L_y = 6.24$ km and $L_z = 1.5$ km. Number of grid cells in spanwise and vertical direction ($N_y^i$ and $N_z^i$) are equal for predictor and optimizer.

| | Optimization horizon $T = 50$ s | | | | Optimization horizon $T = 250$ s | | | | | | | | |
|---|---|---|---|---|---|---|---|---|---|---|---|---|---|
| **Update time** | $T_A = 50$ s | | | | $T_A = 50$ s | | | $T_A = 150$ s | | | $T_A = 250$ s | | |
| $L_x$ [km]: Optimizer | 9.6 | | | | 11.52 | | | 11.52 | | | 11.52 | | |
| Predictor | 10.56 | | | | 12.48 | | | 13.44 | | | 14.4 | | |
| **Grid level $i$** | **0** | **1** | **2** | | **0** | **1** | **2** | **0** | **1** | **2** | **0** | **1** | **2** |
| $N_x^i$: Optimizer | 120 | 160 | 240 | | 144 | 192 | 288 | 144 | 192 | 288 | 144 | 192 | 288 |
| Predictor | 132 | 176 | 264 | | 156 | 208 | 312 | 168 | 224 | 336 | 180 | 224 | 360 |
| $N_y^i$: | 78 | 104 | 156 | | 78 | 104 | 156 | 78 | 104 | 156 | 78 | 104 | 156 |
| $N_z^i$: | 30 | 40 | 60 | | 30 | 40 | 60 | 30 | 40 | 60 | 30 | 40 | 60 |

| | Optimization horizon $T = 150$ s | | | | | | | Optimization horizon $T = 350$ s | | | | | | | | | | |
|---|---|---|---|---|---|---|---|---|---|---|---|---|---|---|---|---|---|---|
| **Update time** | $T_A = 50$ s | | | $T_A = 150$ s | | | | $T_A = 50$ s | | | $T_A = 150$ s | | | $T_A = 250$ s | | | $T_A = 350$ s | |
| $L_x$ [km]: Optimizer | 10.56 | | | 10.56 | | | | 12.48 | | | 12.48 | | | 12.48 | | | 12.48 | |
| Predictor | 11.52 | | | 12.48 | | | | 13.44 | | | 14.4 | | | 15.36 | | | 15.36 | |
| **Grid level $i$** | **0** | **1** | **2** | **0** | **1** | **2** | | **0** | **1** | **2** | **0** | **1** | **2** | **0** | **1** | **2** | **0** | **1** | **2** |
| $N_x^i$: Optimizer | 132 | 176 | 264 | 132 | 176 | 264 | | 156 | 208 | 312 | 156 | 208 | 312 | 156 | 208 | 312 | 156 | 208 | 312 |
| Predictor | 144 | 192 | 288 | 156 | 208 | 312 | | 168 | 224 | 336 | 180 | 240 | 360 | 192 | 256 | 384 | 192 | 256 | 384 |
| $N_y^i$: | 78 | 104 | 156 | 78 | 104 | 156 | | 78 | 104 | 156 | 78 | 104 | 156 | 78 | 104 | 156 | 78 | 104 | 156 |
| $N_z^i$: | 30 | 40 | 60 | 30 | 40 | 60 | | 30 | 40 | 60 | 30 | 40 | 60 | 30 | 40 | 60 | 30 | 40 | 60 |

### 3.1.1 Receding-Horizon Optimal Control Setup

Wind farm operation is optimized over a total horizon of $T_{\text{tot}} = 30$ min, which comprises just under three through-flows for the given wind farm (given a free-stream velocity $U_\infty \approx 9.4$ m s$^{-1}$). In the receding-horizon framework, turbine controls are optimized over windows of horizon $T$ with offset $T_A$ (the control update time) corresponding to the prediction horizon of the predictor. Table 3 summarizes the different combinations of $T$ and $T_A$ considered here. The longest optimization window ($T = 350$ s) allows the optimizer to account for wake interactions over three consecutive rows. With every horizon reduction of 100 s in Tab. 3, the optimizer looses control authority over one row of turbines in the wakes. For $T = 50$ s, wakes cannot propagate to the next row within the optimization window; case 10 may therefore be considered as an 'uncoordinated' control case. Note that optimization and prediction horizons are limited due to the natural divergence of trajectories in chaotic flows. In practice, for our setup, we find that gradients are still accurately represented for $T = 350$ s (the longest optimization horizon considered in this work, see also App. B5 for the gradient verification.





**Table 3.** Optimization horizons $T$ and control update times $T_A$.

| Case | 1 | 2 | 3 | 4 | 5 | 6 | 7 | 8 | 9 | 10 |
|---|---|---|---|---|---|---|---|---|---|---|
| $T$ [s] | 350 | | | | 250 | | | 150 | | 50 |
| $T_A$ [s] | 350 | 250 | 150 | 50 | 250 | 150 | 50 | 150 | 50 | 50 |

**Table 4.** Specifications for the turbine control cases.

| Case | | $C'_{T,\min}$ [$-$] | $C'_{T,\max}$ [$-$] | $\omega_{\max}$ [$^\circ$ s$^{-1}$] |
|---|---|---|---|---|
| Reference | (R) | 2 | 2 | 0 |
| Induction + yaw | (IY) | 0.5 | 2 | 0.4 |
| Steady yaw | (S) | 2 | 2 | 0 |

All cases are initialized from the same flow field depicted in Fig. 7, that was generated by advancing the flow on the reference grid using constant thrust coefficients for all turbines ($C'_T = 2$, no yawing $\omega = 0^\circ$ s$^{-1}$), until a statistically stationary state is achieved. As explained in Sect. 2.5, before each optimization run, the current flow field is taken from the reference simulation
and restricted to the coarser prediction grid and propagated over the control update time $T_A$ in the predictor (Fig. 5). Turbine controls are then optimized over the optimization horizon $T$, starting from initial guess $C'_T = 2$ and $\omega = 0^\circ$ s$^{-1}$ for all turbines, until a stopping criterion is met. The convergence criterion used here is based on the relative improvement of the objective function over the L-BFGS-B iterations, i.e. $\left(\mathcal{J}^{k-1} - \mathcal{J}^k\right)/\mathcal{J}^{k-1} \leq 5 \times 10^{-6}$. The optimized controls are then applied to the reference in the next time window. Since the time-dependent controls are optimized on the temporal grid of the optimizer, the
optimized controls are first interpolated onto the finer temporal reference grid using a simple zero-order hold rule.

### 3.1.2 Turbine Control Cases

Three control scenarios are examined, see Tab. 4. First, we define a steady reference case (R) where turbines operate at Betz-optimal thrust coefficients $C'_T = 2$, aligned with the mean-flow direction ($\theta = 0^\circ$ and $\omega = 0^\circ$ s$^{-1}$). Next, we consider a combined induction and yaw control case (IY) with a maximum yaw rate $\omega_{\max} = 0.4^\circ$ s$^{-1}$. The induction control part is restricted
to the underinduction regime (i.e. $C'_{T,\max} = 2$) to avoid bias in the results due to the inherent inaccuracy of the ADM in the overinduction regime. A response time $\tau = 15$ s is adopted for the time filtering of the thrust coefficient setpoints. Finally, we also consider the steady yaw control case from Sood and Meyers (2022), who used the recursive wake merging methodology from Lanzilao and Meyers (2022) on the Bastankhah wake model (Bastankhah and Porté-Agel, 2016) in a basic optimization framework to determine the yawing setpoints for the TCRWP, subject to a maximum yawing angle of $30^\circ$. For the steady yaw
case here, we simply take their setpoints, and initialize the turbines using these setpoint at the start of the simulation.



## 4 Results and Discussion

This section presents and discusses the results of the optimal control cases. All simulations are conducted on the wICE super-computing platform of the VSC (Vlaams Supercomputer Centrum), using Ice Lake nodes containing 2 Intel Xeon Platinum 8360Y CPUs (36 cores each).

### 4.1 Allocation of resources

The focus of our study is to evaluate the performance of the control models in relation to their computational cost. Computational cost is measured in walltimes, which depend on the number of cores used and the spatial parallelization. For the spatial parallelization, we employ a 2D domain decomposition, similar to the method used in earlier studies involving SP-Wind (see e.g. Goit and Meyers, 2015; Goit et al., 2016; Munters and Meyers, 2018b, 2017).

For each grid resolution, since the time integration in the forward and adjoint simulations is the predominant contribution in the overall walltime, we select the number of compute cores that minimizes the walltime per Runge-Kutta step. Using a maximum of one whole compute node, these scaling tests reveal an optimum of 30, 54 and 72 cores for grid level 0, 1 and 2 respectively. Note that, for every simulation, we reserve the full compute node and then allocate the optimal number of cores using a 'bunch' processor mapping to distribute the cores evenly over both sockets of the Ice Lake node.

### 4.2 Convergence Behavior

Table 5 reports the average number of PDE evaluations (i.e. sum of forward and adjoint simulations) per optimization window required for formal convergence as specified by the convergence criterium from Sect. 3.1.1. Figure 9 shows the L-BFGS-B iterations versus the number of PDE evaluations for the optimization window starting at $t + T_A = 750$ s. To maintain clarity, only cases 4, 7, 9 and 10 (for which $T_A = 50$ s) are depicted in the figure. As expected, the number of PDE evaluations increases with the optimization horizon, as this increases the number of optimization variables. In general, higher resolutions also require more function evaluations.

### 4.3 Power Gains versus Computational Time

Figure 10 shows the performance of the proposed controller versus the computational cost for the three grid resolutions from Tab. 1 and the different combinations of $T$ and $T_A$. Error bars indicating the 95% confidence intervals are also depicted.

The performance of the controllers is measured in terms of the power gains $\eta_P$ and farm efficiency $\eta_{\text{farm}}$:

$$\eta_P = \frac{\overline{P}_{\text{farm}}}{\overline{P}^{\text{ref}}}, \qquad \eta_{\text{farm}} = \frac{\overline{P}_{\text{farm}}}{N_t \overline{P}^{\text{ref}}_{R1}}. \tag{20}$$

The power gain compares the overall power extraction $\overline{P}_{\text{farm}}$ against the power $\overline{P}^{\text{ref}}$ extracted by the Betz-optimal reference case. The farm efficiency evaluates performance against a fictional farm where all turbines operate in the free-stream flow. In that case, the overall power extraction is compared against the average power $\overline{P}^{\text{ref}}_{R1}$ extracted by a Betz-optimal, front-row





**Table 5.** Number of PDE evaluations and L-BFGS-B iterations per optimization run, averaged over the windows, for the different grid resolutions. Standard deviation on number of PDE evaluations is also shown. First window is excluded to account for startup of the controller.

| Case | | 1 | 2 | 3 | 4 | 5 | 6 | 7 | 8 | 9 | 10 |
|---|---|---|---|---|---|---|---|---|---|---|---|
| $T$ [s] | | 350 | | | | 250 | | | 150 | | 50 |
| $T_A$ [s] | | 350 | 250 | 150 | 50 | 250 | 150 | 50 | 150 | 50 | 50 |
| **L-BFGS-B iterations** | level 0 | 27.6 | 26.3 | 22.7 | 23.8 | 19.3 | 17.8 | 17.9 | 10.2 | 11.0 | 8.1 |
| | level 1 | 28.2 | 27.6 | 25.2 | 24.6 | 20.9 | 20.5 | 18.7 | 11.1 | 10.6 | 7.9 |
| | level 2 | 31.8 | 32.7 | 29.9 | 30.3 | 23.6 | 22.7 | 21.1 | 11.8 | 11.4 | 8.1 |
| **PDE evaluations** | level 0 | 58.2 | 55.6 | 48.5 | 50.7 | 41.6 | 38.6 | 38.9 | 23.5 | 25.1 | 19.2 |
| | level 1 | 59.4 | 57.9 | 53.4 | 52.2 | 44.7 | 43.9 | 40.5 | 25.2 | 24.2 | 18.9 |
| | level 2 | 66.6 | 68.4 | 62.8 | 63.7 | 50.1 | 48.5 | 45.3 | 26.6 | 25.9 | 19.1 |
| **Standard deviation** | level 0 | 4.1 | 6.0 | 4.5 | 5.9 | 3.2 | 4.2 | 3.6 | 4.0 | 4.3 | 1.3 |
| | level 1 | 3.8 | 5.9 | 4.7 | 6.2 | 6.0 | 5.3 | 4.2 | 4.2 | 4.1 | 1.4 |
| | level 2 | 3.8 | 4.6 | 5.5 | 7.2 | 4.0 | 3.6 | 4.9 | 4.4 | 4.7 | 0.8 |

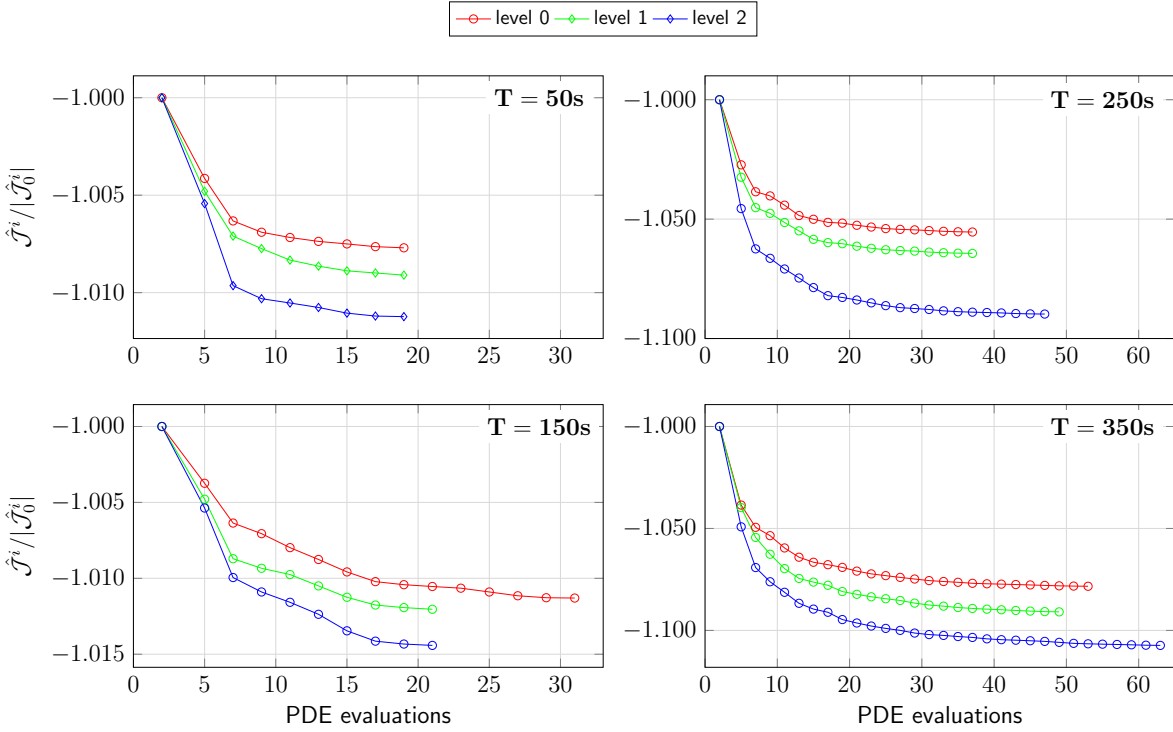

**Figure 9.** L-BFGS-B iterations for cases 4, 7, 9 and 10 (with $T_A = 50$ s) for the different grid levels for the optimization window starting at $t + T_A = 750$ s. For every grid level $i$, the objective function $\hat{\mathcal{J}}^i$ is scaled by $\hat{\mathcal{J}}_0^i$ of the first iteration.





reference turbine. For the power computations, we only consider turbine operation after $t_{\mathrm{startup}} = 300$ s to take into account the startup time of the controllers. Error bars on $\eta_P$ and $\eta_{\mathrm{farm}}$ are computed starting from $t_{\mathrm{startup}} = 300$ s using block bootstrapping with window length $600$ s.

Computational times are measured in terms of the real-time factor

$$RT = \frac{\bar{t}_{\mathrm{wall}}}{T_A}, \tag{21}$$

where $\bar{t}_{\mathrm{wall}}$ is the walltime per optimization run, averaged over all optimization windows of the corresponding case. Error bars on the real-time factor are based on the deviations of the walltimes over the different optimization windows. In the time-delayed MPC loop from Fig. 2(**b**) and Fig. 3, for real-time operation, all computations for a given receding-horizon window should be performed within a time interval of length $T_A$ (the control update time). In the present study, we only consider the computational time for the optimization and omit details of the state estimation. In practice, depending on the method, estimation may take as long as the optimization of the controls, such that $RT < 0.5$ is expected to be sufficiently fast for real-time operation. Note that the flow prediction subsequent to the state estimation (cf. Fig. 3) corresponds to one forward simulation on the coarse grid; computational time for the prediction is hence negligible compared to that of optimization (and possibly estimation).

### 4.3.1 Analysis of Computational Cost

First of all, as can be appreciated in Fig. 10, all real-time factors are bigger than one, ranging from 1.79 to 270. Three observations can be made: the real-time factor and hence the computational cost of the optimization increases if (**a**) the optimization horizon $T$ increases, (**b**) the update time $T_A$ is reduced, or (**c**) the grid is refined. Case 10 on grid level 0 — with $T = 50$ s, $T_A = 50$ s and the lowest number of grid points of all simulations — is therefore the fastest and only a factor 1.79 (on average) slower than real-time. Conversely, case 0 on grid level 2 is the most challenging in terms of walltime, with a real-time factor of 270. In terms of $RT$, Fig. 10 reveals that, in general, the relative order of the cases remains unchanged when refining the grid.

### 4.3.2 Analysis of Power Gains and Farm Efficiency

The power gains from Fig. 10 are more clearly summarized in Fig. 11. As can be expected for the fully aligned farm layout, the farm efficiency for the uncontrolled Betz-optimal reference case is relatively low at approximately $45$ %. From Fig. 10 and Fig. 11, it can be seen that all optimal control cases improve on the uncontrolled reference, except for case 10 due to its short optimization horizon ($T = 50$ s, $T_A = 50$ s). The highest power gain is observed for case 4 ($T = 350$ s, $T_A = 50$ s) on grid level 1: $\eta_P = 1.51$. Two main trends can be observed: (**a**) increasing the optimization horizon $T$ increases the power extraction, and (**b**) decreasing the control update time $T_A$ increases the power extraction.

Interestingly, the grid resolution has no clear impact on the power gains. In some cases, refining the grid increases the power extraction, sometimes the power extraction decreases. However, the performance gains and losses that come from refining or coarsening the grid (for a given $T$ and $T_A$) are only marginal compared to the effects of changing $T$ and $T_A$ and mostly fall within the bounds of the confidence interval. Overall, the influence of the receding-horizon parameters is hence much bigger than that of the grid resolution.

**Figure 10.** Gain factor (left axis) and farm efficiency (right axis) versus real-time factor for the optimal control cases, including error bars. Betz-optimal reference case and steady yaw control case are also indicated.

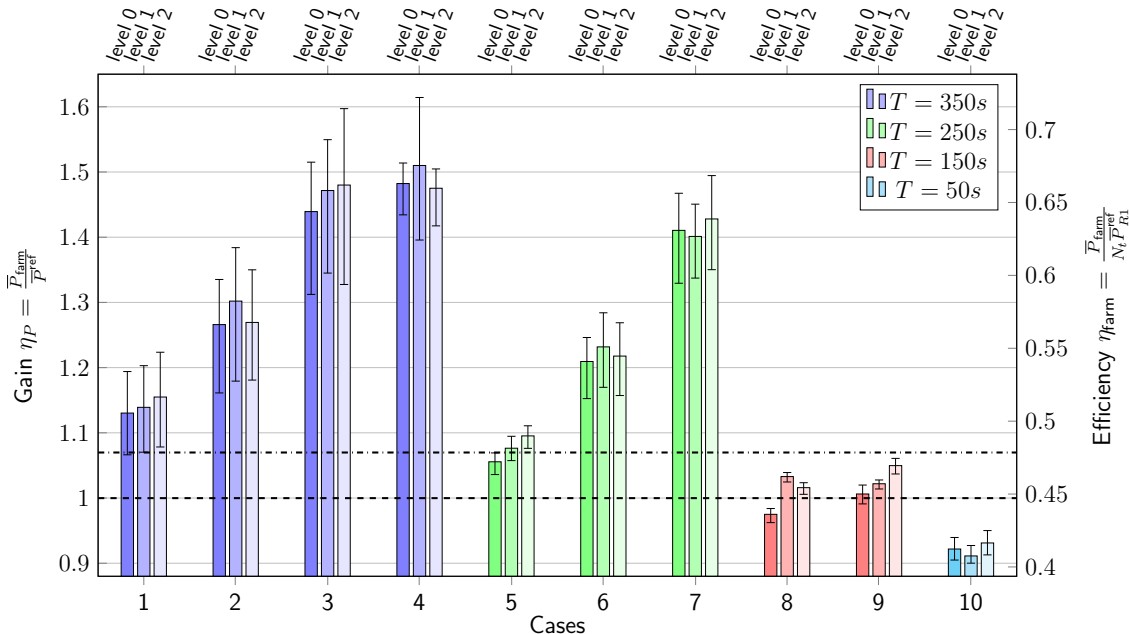

**Figure 11.** Gain factor (left axis) and farm efficiency (right axis) for the optimal control cases, including error bars. Betz-optimal reference case and steady yaw control case are also indicated (dotted lines).

### 4.4 Yaw and Induction Characteristics

Figure 12 and Fig. 13 illustrate the time evolution of respectively the filtered thrust coefficients $\hat{C}'_T$ and yaw angles $\theta$ for the eight turbines in column C1 (see Fig. 6) for optimal control cases 1, 4, 5, 7, 8, 9 and 10 after optimization on grid level 0. We only show results for column C1, since the observations for the other turbine columns are similar. The thrust coefficients and yaw angles for the other grid levels are shown in App. C, but the trends observed there are similar as the ones for grid level 0.

For cases 8–10, characterized by the shortest time horizons $T$, wakes cannot propagate from one row of turbines to the next row within the optimization window. In those cases, power is therefore maximized at the level of individual turbines: all turbines operate at the Betz limit $\hat{C}'_T = 2$ while oscillating around the flow-aligned yaw angle of $0°$. The magnitude of the oscillations increases for downstream turbines to account for the higher flow angles that exist in downstream regions of the wind farm due to the unsteadiness in the local flow. For these cases, the control update time $T_A$ has no impact on the controls.

As the time horizon increases, significant yaw angles emerge for upstream turbines, with even some quasi-static yawing behavior for the cases with short update times. This is particulary evident for case 4 ($T = 350$ s, $T_A = 50$ s), where front-row turbines are immediately redirected to a yaw angle of $\pm 30°$. Downstream turbine rows 2–4 exhibit similar behavior, but at lower misalignment angles and with more complex oscillations in response to the local unsteadiness of the flow. In contrast, the last few rows again operate around the unyawed position. Note that the misalignment angles of $\pm 30°$ for the front-row turbines matches the value obtained using the static yaw controller from Sood and Meyers (2022). However, the





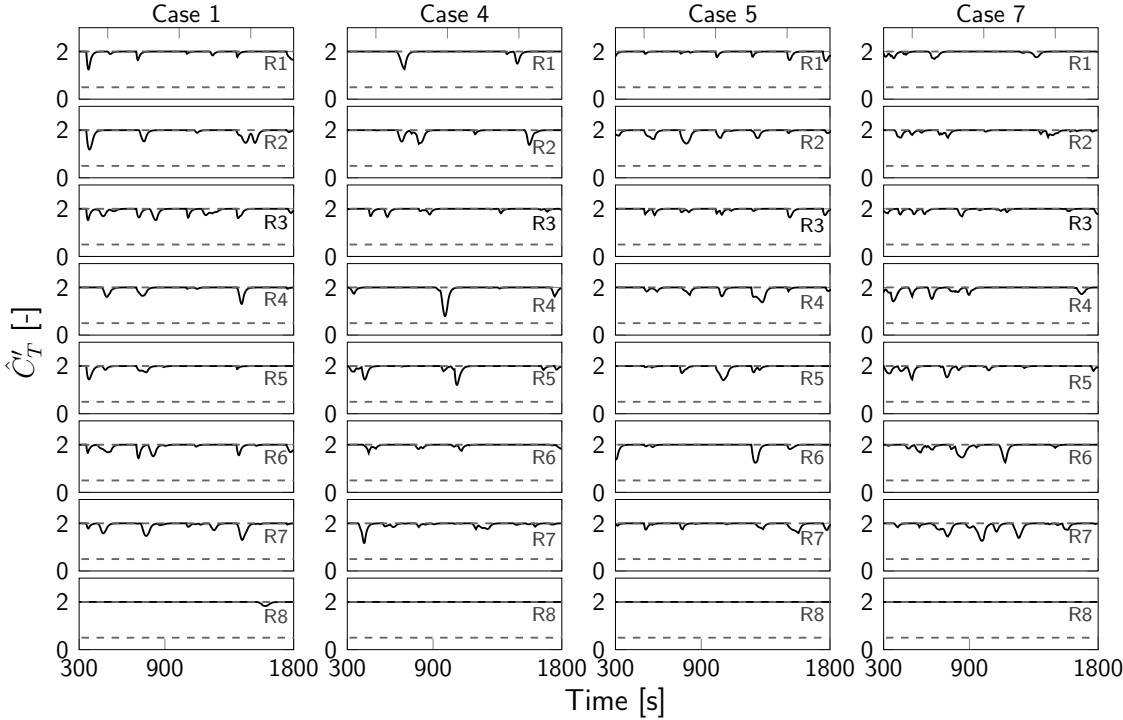

**Figure 12.** Time evolution of filtered thrust coefficients $\hat{C}'_T$ for turbine column C1 for different optimal control cases for grid level 0. For cases 8–10 (not shown), the thrust coefficients do not deviate from the Betz-optimal value $\hat{C}'_T = 2$.

main difference compared to the steady yaw case is the distinct yawing of downstream turbines (starting in row 2 already), resulting in significant gains. Case 7 behaves similar to case 4 in terms of yawing, but the quasi-static yawing is less pronounced

due to the shorter optimization horizon. Also note that transitions of the yaw angle between $-30°$ and $30°$, as observed for front-row turbines for case 4 and case 7, is propagated in the mean flow to downstream turbines, as indicated by the red line in Fig. 13.

   Increasing the control update time $T_A$ (see e.g. case 1 and 5) eliminates the quasi-static yawing behavior observed in the upstream turbines (as discussed above for case 4). This is because the controller redirects yawed turbines to the unyawed

position at the end of the optimization window, as the corresponding wakes cannot propagate to the next turbine row anymore, rendering yaw control disadvantageous. This end-of-time effect is detrimental to the long-term power extraction, since it is merely an artefact of the finite optimization horizon. For case 4 ($T_A = 50$ s, and analogeously for case 7), the end-of-time effect is mitigated by the shorter control update time ($T_A \ll T$). In contrast, for case 1 ($T_A = 350$ s), upstream turbines are steered towards a yawed position at the start of every optimization window, but then redirected at the end of the window due to

the end-of-time effect. This process repeats for the consecutive windows. Case 5 exhibits similar behavior, but the time spent in the yawed position is shorter due to the shorter optimization window (a larger portion of the windows is affected by the



**Figure 13.** Time evolution of yaw angles $\theta$ for turbine column C1 for different optimal control cases for grid level 0.





end-of-time effect). Case 2 and 3, and case 6 (not shown in Fig. 12 and Fig. 13) display behavior similar to cases 1 and 5 respectively, but the end-of-time effect is less pronounced due to the shorter control update time.

Overall, yaw control emerges as the dominant control mechanism for the TotalControl wind farm. Only for longer opti-
mization horizons ($T \geq 250\,\mathrm{s}$), induction control is used resulting in minor deviations from the Betz-optimal thrust coefficient. Interestingly, for cases 1, 4, 5 and 7 and for the front-row turbines, it seems that the dips in $C_T$ for the upstream turbines roughly coincide with the zero-crossings of the yaw angle of those turbines. This shows the connection between yaw and induction control: for the given setup, yaw control is the dominant control mechanism, and induction control is used as an additional control mechanism when there is no (instantaneous) yawing. Simulations (not shown here) suggest that yaw control only (i.e.
disabling induction control) does not entail a significant performance reduction.

Remark that the discussion in this section was limited to the effect of the receding-horizon parameters on the thrust coeffi-
cients and yaw angles. More general aspects of wind farm control — such as the curtailing of power of front-row turbines in favor of downstream turbines, the typical absence of yawing for downstream turbines etc. — are similar to previous wind farm control studies such as the ones in Munters and Meyers (2017, 2018b), the reader is referred there for a more fundamental view
on the physics of wind farm control.

### 4.5 Discussion

Figure 11 clearly shows that all optimal control cases outperform the Betz-optimal reference case in terms of power gain, except case 10 (due to its short optimization horizon $T = 50\,\mathrm{s}$). If the time horizon is long enough, i.e. $T > 150\,\mathrm{s}$ (cases 1–7), we also obtain significant improvements over the steady yaw controller from Sood and Meyers (2022). This is a non-trivial
observation, given the coarseness of the control models and the accompanying model mismatch compared to the fine-grid reference, especially in context of the time-decoupled MPC loop including the prediction step. This suggests that large-scale structures (in space and time) in the wind farm boundary layer may already suffice to extract an adequat control signal. Short-term evolutions of the boundary layer are not accurately captured by the coarser control models, resulting in only limited gains (for cases 8 and 9) or even losses (case 10) compared to the Betz-optimal reference. Extending the time horizon allows the
optimizer to tailor the controls based on the large-scale spatio-temporal structures, that are described sufficiently accururately by the control model, hence resulting in significant improvements. It must also be noted that increasing the horizon (either through $T$ or $T_A$) increases the variability on the results, resulting in larger error bars on the power gains (compare e.g. the error bars of cases 1–4 to those of cases 8–10). Furthermore, increasing $T_A$ entails performance losses not only due to the end-of-time effect, but also due to the increased prediction error in the predictor.
Interestingly, refining the optimization resolution does not significantly improve the results in terms of power gain, and can even be disadvantageous in some cases (at least in the range of resolutions considered in this paper). This effect may be attributed to the model mismatch: upon refining, the controls are tuned to account for the additional small(er)-scale variations, but these (incremental) adjustments are not necessary optimal on the fine-grid reference. In Bauweraerts and Meyers (2019) (in the context of turbulent forecasting), it was shown that modeling errors even slightly decrease with grid coarsening, due to
the decreasing subgrid-scale errors in the LES and the decreased effect of chaotic divergence of solution trajectories on coarser





grids. In other words: it may be better to only optimize for the large scales than to also take into account smaller scales that may be inaccurately modeled. However, it must be noted that even the finest optimization resolution (level 2) is still more than three times coarser than the reference resolution. It is expected that further grid refinements would eventually improve on the coarse grid results, when the actual small-scale variations are sufficiently accuratley described by the control models. However,

these kind of similations would be prohibitive due to excessive computational costs.

Finally, it must be noted that the coarse control models are incapable of capturing all the turbulent dynamics governing the optimal wind farm control problem. On the one hand, it seems that they can accurately model large-scale motions in the flow, such as the general deflection of the wake under yawed conditions and the gross behavior of the wakes. On the other hand, the shedding of vortex rings, that play a crucial role in enhancing wake mixing when dynamically controlling turbine thrust

(Munters and Meyers, 2018a), cannot be represented on coarse grids. As such, the proposed methodology is less eligible for dynamic induction control, and yaw control emerges as the dominant mechanism for coarse-grid LES-based control, where in that case the gains mainly originate from dynamically steering away the wakes from downstream turbines, synchronized to the turbulent inflow.

## 5   Towards real-time Optimal Wind Farm Control

The discussion in Sect. 4.3 has revealed that, on the one hand, the optimization horizon $T$ needs to be long enough to take into account wake interactions over subsequent turbine rows. On the other hand, the control update time needs to be short enough to discard end-of-time effects. Furthermore, it was shown that refining the grid resolution significantly increases the computational cost without a signficant improvement in the performance of the controller in terms of power gain. By leveraging these insights, we now design a competitive (in terms of power gains) controller that is as close to real-time as possible.

### 5.1   Case setup

For the receding-horizon parameters, we propose $T = 300$ s and $T_A = 120$ s. An optimization horizon of $T = 300$ s allows the controller to account for wake interactions during the optimization. With $T_A = 120$ s, the final $180$ s of the optimization window are discarded, which is roughly the portion of the window affected by end-of-time effects. Based on the analysis in Sect. 4.3, longer update times are expected to reduce performance due to end-of-time effects. Although further decreasing $T_A$

could potentially improve the power gains by mitigating the model mismatch, the benefits would be relatively insignificant compared to those that come from addressing the end-of-time effect, and would hence result in an undue increase in the real-time factor. Regarding the convergence criteria, we consider three cases: case (**a**) that uses the same convergence criterium as in Sect. 3.1.1 (i.e. $\left(\mathcal{J}^{k-1} - \mathcal{J}^k\right)/\mathcal{J}^{k-1} \leq 5 \times 10^{-6}$), and cases (**b**) and (**c**) that additionally impose a maximum number of optimization iterations, respectively $N_{\text{opt,max}} = 10$ and $N_{\text{opt,max}} = 5$. Furthermore, given the limited contribution of induction

control in the overall power gains, all thrust coefficients are kept constant to the Betz-optimal value $C_T' = 2$. By disabling induction control, the number of optimization variables decreases by a factor 2, which may potentially speed up the convergence of the optimization problems.



**Table 6.** Case description for the case from Sect. 5.

| Simulation setup | | | | Remarks |
|---|---|---|---|---|
| Space resolution [m$^3$] | $\Delta x \times \Delta y \times \Delta z$ | $80 \times 80 \times 50$ | | Grid level 0 |
| Time resolution [s] | $\Delta t$ | 4 | | Coarser compared to Tab. 1 |
| Grid cells | $N_x \times N_y \times N_z$ | $144 \times 52 \times 15$ | | |
| Domain size [km$^3$] | $L_x \times L_y \times L_z$ | $11.52 \times 4.16 \times 0.75$ | | Cropping in y- and z-direction compared to Tab. 2 |
| Pressure grad [m/s$^2$] | $\partial_x p_\infty / \rho$ | $1.045 \times 10^{-4}$ | | Changed due to cropping in z-direction compared to Tab. 2 |

| Receding-horizon parameters | | | | Remarks |
|---|---|---|---|---|
| Horizon [s] | $T$ | 300 | | |
| Update time [s] | $T_A$ | 125 | | |
| L-BFGS-B iterations | $N_{\text{opt,max}}$ | / | | Case **(a)** |
| | | 10 | | Case **(b)** |
| | | 5 | | Case **(c)** |

**Table 7.** Gain factor versus real-time factor for the optimal control cases. Error bars are also shown in the brackets.

| | Case **(a)** | Case **(b)** | Case **(c)** |
|---|---|---|---|
| Gain $\eta_P$ | 1.39 [0.09,0.07] | 1.38 [0.07,0.05] | 1.31 [0.04,0.03] |
| Real-time factor | 4.08 [0.14,0.13] | 2.33 [0.002,0.002] | 1.33 [0.001,0.002] |

To further minimize the real-time factor, simulations are performed on the coarsest grid level (level 0). We use the streamwise
domain length from the cases with horizon $T = 250$ s (cf. Tab. 2): $L_x = 11.52$ km. Earlier results (not shown) suggest that
this domain is still big enough to prevent recycling of the wakes into the front-row turbines. Furthermore, in comparison to
Table 2, we apply an additional cropping of the optimization domain in the spanwise and vertical directions to further reduce
computational costs, resulting in $L_y = 4.16$ km and $L_z = 0.75$ km. To keep a friction velocity of $u_\tau = 0.28$ m s$^{-1}$ for the new
vertical dimension, the pressure gradient is adjusted to $\partial_x p_\infty / \rho = 1.0453 \times 10^{-4}$ m s$^{-2}$. For the time step, we take $\Delta t = 4$ s,
which is the coarsest time step that still respects a CFL number of $0.8$. The simulation setup is summarized in Table 6.

**5.2 Results and discussion**

The results for cases **(a)**, **(b)** and **(c)** are summarized in Tab. 7. By tailoring the control model and through a sensible choice
for the receding-horizon parameters as described above, for case **(c)** we achieve a real-time factor of 1.33 with only a minor
decrease in power gain compared to cases **(a)** and **(b)** (1.31 versus 1.39 and 1.38 respectively). Apparently, just 5 L-BFGS-B
iterations already suffice to extract an adequate control action.





It is important to note that the setup used here was tailored to the fully aligned TCRWP, and therefore the results may not directly translate to other farm configurations. Nevertheless, the observations suggest that using coarse-grid LES for real-time wind farm control is a viable approach, if another order of magnitude (factor 10) in computational or algorithmic speed-up can be found (since, for practical wind farm control, the optimization process must be at least twice as fast as real-time to perform state estimation within the computational window). However, as CPU computing power continues to advance, and

since this first investigation already achieves near real-time speed, a factor 10 still remains within reach. Further speed-ups may also be attained through GPU-accelerated computing. Furthermore, it is worth mentioning that the SP-Wind code can also be enhanced. For instance, a new version of the code is currently being developed that employs 3D domain decomposition for the spatial parallelization (as opposed to the 2D domain decomposition used in this work). This upgrade will be necessary to handle even bigger optimization cases in real-time. Very recently, Janssens and Meyers (2023) proposed a multiple shooting

algorithm for large-scale optimal control cases, such as the ones considered here. The additional speed-up due to the temporal parallelization in that case may potentially narrow the gap towards achieving actual, practical wind farm control in real-time.

## 6   Conclusions

In the current manuscript, we investigated the influence of the grid resolution of the LES-based control model and receding-horizon parameters on the performance of the controller, both in terms of power gain and computational cost. To that end, we

defined a set of optimal control cases with varying optimization horizons and control update times, as well as a fine-grid LES emulator model, applied to the TotalControl Reference Wind Power Plant. For each case, we defined three grid resolutions for the LES-based control model, and performed a complete optimal control simulation on each of the grids.

Regarding the receding-horizon parameters, on the one hand the results indicate that the optimization horizon should be long enough to take into account turbine-wake interactions over subsequent turbine rows. In that case, upstream turbines

are misaligned to steer away the wakes from downstream turbines in a quasi-static way, resulting in significant power gains compared to the uncontrolled reference case and steady yaw control case. Downstream turbines are yawed as well, but to a lesser extent and in a more complex pattern, mostly in response to the local unsteadiness of the flow. On the other hand, the control update times should be short enough to mitigate end-of-time effects, i.e. to discard controls near the end of the optimization window that are affected by the finiteness of the optimization window. Taking this into account, optimal control

case 4 (with $T = 350$ s and $T = 50$ s, i.e. the longest horizon and shortest update time), consistently produce the highest power gains, up to $\eta_P = 1.51$ on grid level 1. Furthermore, it must be noted that all optimal control cases improve on the uncontrolled reference case in terms of power gain, except case 10 ($T = 50$ s, $T_A = 50$ s) due to the short optimization horizon. Moreover, cases 1–7 (where $T \geq 250$ s) also outperform a steady yaw control case that was optimized based on the Bastankhah wake model combined with a recursive wake merging merhod (see Sood and Meyers, 2022). Obviously, increasing the optimization

horizon and decreasing the control update time both increase the real-time factor.

Somewhat surprisingly, the grid resolution has no significant impact on the performance of the controllers, at least not in the range of resolutions considered here. Sometimes refining the grid resolution results in higher gains, sometimes the gains





decrease. This may be attributed to the existence of many local optima in the wind farm optimization problem, as well as the model uncertainty, since the finest grid in this work was still more than three times coarser than the reference simulation. It

is expected that finer grids (with resolutions close to that of the reference), would eventually improve on the coarse models investigated here, as soon as the small scales are represented sufficiently accurately to take them into account in the control action. However, optimizations on these kind of resolution would be prohibitive due to computational cost and outweigh the potential gains originating from the decreased model mismatch. It must also be noted that we did not investigate the influence of the convergence criterion, therefore it is possible that somewhat better results can be obtained on each of the grid levels by

tweeking the stopping criterium.

In terms of power gains, the coarse grid control models investiagetd in the current manuscript perform surprisingly well, with gains up to and above $40\%$ if the time horizon is sufficiently long and the update time sufficiently short. This is a powerful result, given the model uncertainty due to the additional prediction step in the time-delayed MPC loop. Regarding the complex physics underlying wind farm control, the results suggest that the large-scale spatial and temporal structures in the wind farm

boundary layer (i.e. the ones that are accurately represented by the coarse LES models), suffice to extract an efficient yaw controller. From the viewpoint of real-time LES-based optimal control, this means that high-performance controllers can be obtained at only a fraction of the computational cost by coarsening the grid resolution, potentially bridging the gap between theoretical studies based on LES and practical, real-time wind farm control. Via a proper choice of the receding-horizon parameters and optimal control domain, we achieved a gain of 31 % with real-time factor 1.33, i.e. only 1.33 times slower than

real-time. Even better control signals (in terms of the resulting power gains) may be obtained through an enhanced wake mixing by better exploiting dynamical induction control (Munters and Meyers, 2018a) or via the helical wake structures originating from individual pitch control (Frederik et al., 2020). However, this would require a much finer numerical grid, which currently does not allow for a real-time implementation in LES.

It must be noted that, upon the restriction from the actual wind farm (the fine-grid LES wind farm model) to the coarse res-

olution of the prediction and control model by the restriction operator, we still assume the entire flow field from the reference simulation is available. In practice, this is not the case; the flow field is only available in the form of discrete measurements, for example LiDAR measurements or SCADA data. Consequently, for a practical controller, the flow field must first be reconstructed from these measurements in a state estimator, for example using Kalman filtering or 4D variational data assimilation (see e.g. Bauweraerts and Meyers, 2021). Future work should therefore focus on the design of an efficient state estimator

tailored to wind farm control, which can then be incorporated into the LES-based wind farm control methodology proposed here. As shown in previous work from Bauweraerts and Meyers (2019), LES-based state estimation may potentially benefit from grid coarsening as well. However, in that case, the question still remains whether combined LES-based state estimation and control can be fast enough for real-time applications, and how the additional reconstruction errors originating from the state estimation will affect the performance of the controllers. This would be the next step towards actual, practical wind farm

control.

We also admit that the considered test case (TCRWP) is quite susceptible to high power gains due to the fully aligned turbine configuration. Applying the proposed wind farm controller to other wind farm layouts could be a topic for future





**Table A1.** Look-up table for grid level 0 ($\Delta x \times \Delta y \times \Delta z = 80 \times 80 \times 50$ m$^3$): correction factor $M$ from eq. (10) for different thrust coefficient $\hat{C}'_T$ and yaw angle $\theta$.

| | | $\hat{C}'_T$ | | | | | |
|---|---|---|---|---|---|---|---|
| | | 0.5 | 1.0 | 1.5 | 2.0 | 2.5 | 3.0 |
| | 0° | 0.9498 | 0.9020 | 0.8571 | 0.8147 | 0.7752 | 0.7385 |
| | 5° | 0.9499 | 0.9023 | 0.8576 | 0.8156 | 0.7763 | 0.7398 |
| | 10° | 0.9502 | 0.9037 | 0.8593 | 0.8180 | 0.7794 | 0.7434 |
| | 15° | 0.9508 | 0.9057 | 0.8620 | 0.8218 | 0.7843 | 0.7492 |
| $\theta$ | 20° | 0.9516 | 0.9067 | 0.8654 | 0.8267 | 0.7906 | 0.7567 |
| | 25° | 0.9525 | 0.9102 | 0.8694 | 0.8323 | 0.7978 | 0.7655 |
| | 30° | 0.9536 | 0.9129 | 0.8734 | 0.8381 | 0.8057 | 0.7752 |
| | 35° | 0.9549 | 0.9158 | 0.8786 | 0.8440 | 0.8137 | 0.7852 |
| | 40° | 0.9563 | 0.9179 | 0.8838 | 0.8508 | 0.8222 | 0.7952 |

research, although it is expected that the controller will still be able to produce competitve gains. To increase the credibility of the proposed control strategy in general, another interesting direction would be to use more accurate turbine models in the
wind farm emulator model (e.g. actuator-sector models (ASM) or actuator-line models (ALM)), instead of the very basic non-rotating, actuator-disc model used here. In the first place, one could investigate the effect of the additional model mismatch when using an ADM control model in the coarse LES in combination with an ASM or ALM reference model.

## Appendix A: Look-up Tables for Power Correction

This appendix lists the correction factors $M$ from eq. (10) for the disc-averaged velocity for different values of the thrust
coefficient $\hat{C}'_T$ and yaw angle $\theta$ for the three grid resolutions considered in the present manuscript. The corrections are chosen is such a way that the corrected disc-averaged velocity on the coarse grid equals the disc-averaged velocity on the reference grid for a given thrust coefficient and yaw angle for uniform inflow. The resulting look-up values are summarized in Tab. A1, Tab. A2 and Tab. A3 for grid level 0, grid level 1 and grid level 2 respectively. These tables are constructed based on uniform inflow simulations with $U_\infty = 8$ m s$^{-1}$, prescribed using a fringe region spanning the final 20 % of the simulation domain. All
simulations are conducted for a single DTU 10MW reference turbine using the same domain size $L_x \times L_y \times L_z = 26.92D \times 13.46D \times 8.41D$, where $D = 178.3$ m is the rotor diameter. For the resolution of the reference, we use $\Delta x \times \Delta y \times \Delta z = 13.33 \times 13.33 \times 6.67$ m$^3$ as prescribed by Tab. 1. For a given $\hat{C}'_T$ and $\theta$, the corresponding correction factor $M$ is computed by (linearly) interpolating the look-up tables.





**Table A2.** Look-up table for grid level 1 ($\Delta x \times \Delta y \times \Delta z = 60 \times 60 \times 37.5$ m$^3$): correction factor $M$ from eq. (10) for different thrust coefficient $\hat{C}'_T$ and yaw angle $\theta$.

|  |  | $\hat{C}'_T$ |  |  |  |  |  |
|---|---|---|---|---|---|---|---|
|  |  | 0.5 | 1.0 | 1.5 | 2.0 | 2.5 | 3.0 |
|  | 0° | 0.9596 | 0.9205 | 0.8834 | 0.8468 | 0.8125 | 0.7803 |
|  | 5° | 0.9597 | 0.9208 | 0.8834 | 0.8476 | 0.8136 | 0.7816 |
|  | 10° | 0.9601 | 0.9217 | 0.8850 | 0.8500 | 0.8167 | 0.7852 |
|  | 15° | 0.9606 | 0.9240 | 0.8874 | 0.8535 | 0.8213 | 0.7907 |
| $\theta$ | 20° | 0.9612 | 0.9247 | 0.8904 | 0.8578 | 0.8269 | 0.7974 |
|  | 25° | 0.9621 | 0.9281 | 0.8938 | 0.8627 | 0.8332 | 0.8051 |
|  | 30° | 0.9630 | 0.9290 | 0.8974 | 0.8678 | 0.8398 | 0.8133 |
|  | 35° | 0.9641 | 0.9313 | 0.9011 | 0.8730 | 0.8466 | 0.8217 |
|  | 40° | 0.9652 | 0.9337 | 0.9048 | 0.8781 | 0.8532 | 0.8297 |

**Table A3.** Look-up table for grid level 2 ($\Delta x \times \Delta y \times \Delta z = 40 \times 40 \times 25$ m$^3$): correction factor $M$ from eq. (10) for different thrust coefficient $\hat{C}'_T$ and yaw angle $\theta$.

|  |  | $\hat{C}'_T$ |  |  |  |  |  |
|---|---|---|---|---|---|---|---|
|  |  | 0.5 | 1.0 | 1.5 | 2.0 | 2.5 | 3.0 |
|  | 0° | 0.9709 | 0.9430 | 0.9141 | 0.8869 | 0.8606 | 0.8354 |
|  | 5° | 0.9709 | 0.9423 | 0.9144 | 0.8873 | 0.8611 | 0.8360 |
|  | 10° | 0.9711 | 0.9427 | 0.9152 | 0.8884 | 0.8625 | 0.8376 |
|  | 15° | 0.9714 | 0.9434 | 0.9163 | 0.8902 | 0.8648 | 0.8403 |
| $\theta$ | 20° | 0.9716 | 0.9455 | 0.9180 | 0.8926 | 0.8680 | 0.8440 |
|  | 25° | 0.9723 | 0.9460 | 0.9194 | 0.8954 | 0.8718 | 0.8488 |
|  | 30° | 0.9728 | 0.9481 | 0.9219 | 0.8985 | 0.8760 | 0.8543 |
|  | 35° | 0.9734 | 0.9477 | 0.9241 | 0.9016 | 0.8802 | 0.8599 |
|  | 40° | 0.9741 | 0.9491 | 0.9263 | 0.9048 | 0.8845 | 0.8772 |

## Appendix B:  Derivation and Verification of the temporally discrete Adjoint Method and Gradient

In this Appendix, we formulate the temporally discrete adjoint method and derive expression (18) for the cost functional gradients. These gradients are also validated via finite differences. The derivation and notation are similar to the one from Yilmaz and Meyers (2019), differences are explicitly formulated throughout the text.



## B1   Discretization of the Optimal Control Problem

The discrete adjoint method first discretizes and then linearizes the state equations, next it formulates the discrete adjoint of the linear system (Giles and Pierce, 2000). Using an explicit fourth order Runge–Kutta discretization with $N$ time steps, the discretization of the optimization problem (1)–(7) can symbolically be written as

$$\min_{\boldsymbol{\varphi},\boldsymbol{q}} \tilde{\mathcal{J}}^N = \sum_{n=1}^N I^n \qquad \text{s.t.} \qquad \boldsymbol{q}_i^n = \boldsymbol{q}^n + \Delta t \sum_{j=1}^4 a_{ij} \boldsymbol{Y}(\boldsymbol{q}_j^n, \varphi^n) \qquad i = 1\ldots 4, \tag{B1}$$

$$\boldsymbol{q}^{n+1} = \boldsymbol{q}^n + \Delta t \sum_{i=1}^4 b_i \boldsymbol{Y}(\boldsymbol{q}_i^n, \varphi^n), \tag{B2}$$

where $\boldsymbol{q}^n = [\boldsymbol{u}^n, p^n, \hat{C}_T'^n, \theta^n]$ and $\varphi^n = [C_T'^n, \omega^n]$ are respectively the state variables and controls at time instant $t^n = n\Delta t$. In the Runge–Kutta stage equations, $\boldsymbol{Y}$ denotes the right-side of the governing equations consisting of the Navier-Stokes equations, thrust coefficient filter equation and yaw rate equation. $I_n$ denotes the discretized objective functional. Below, we continue the derivation for a general explicit $(j < i)$ fourth-order Runge–Kutta scheme; the simulations in SP-Wind are carried out using classical Runge–Kutta 4, for which the nonzero Butcher tableau coefficents are $a_{21} = a_{32} = 1/2$, $a_{43} = 1$, $b_1 = b_4 = 1/6$ and $b_2 = b_3 = 1/3$.

As opposed to the basic first-order discretization from Yilmaz and Meyers (2019), in this work the intermediate Runge–Kutta stages are also used in the discretization of the cost functional. In order to achieve this, the continuous cost functional (1), here symbolically written as $\tilde{\mathcal{J}}(\varphi, \boldsymbol{q}(\varphi)) = \int_0^T J(\varphi, \boldsymbol{q}(\varphi)) dt$, is first rewritten in the form of an ordinary differential equation (ODE),

$$\frac{d\mathcal{J}_t(t)}{dt} = J(\varphi(t), \boldsymbol{q}(t)) \qquad \mathcal{J}_t(0) = 0, \tag{B3}$$

where $\tilde{\mathcal{J}}(\varphi, \boldsymbol{q}(\varphi)) = \mathcal{J}_t(T)$. The ODE in (B3) is then discretized using Runge–Kutta 4, yielding a new expression for the discretized cost function that is of fourth-order accuracy:

$$\tilde{\mathcal{J}}^N = \sum_{n=1}^N I^n = \sum_{n=1}^N \left( \Delta t \sum_{i=1}^4 b_i J(\varphi^n, \boldsymbol{q}_i^n) \right). \tag{B4}$$

Remark that compared to Yilmaz and Meyers, the control variables $\varphi^n$ in the Runge–Kutta discretization in (B1)–(B2) are kept constant over the Runge–Kutta stages, since (in practice) controls are kept constant over the control time step (which we assume here is equal to the discretization time step $\Delta t$).

## B2   Linearization of the State Equation

Analogeously to Yilmaz and Meyers, the discretized state equations in (B1)–(B2) can be linearized, resulting in a linear system for every time step $n$:

$$\mathsf{K}^n \mathsf{L}^n = \mathsf{M}^n, \tag{B5}$$





590

$$
\mathrm{K} = \begin{bmatrix}
-1 & 0 & 0 & 0 & 0 & 0 \\
1 & -1 & 0 & 0 & 0 & 0 \\
1 & a_{21}\Delta t \mathrm{Y}_{\boldsymbol{q}}(\boldsymbol{q}_1^n,\varphi^n) & -1 & 0 & 0 & 0 \\
1 & a_{31}\Delta t \mathrm{Y}_{\boldsymbol{q}}(\boldsymbol{q}_1^n,\varphi^n) & a_{32}\Delta t \mathrm{Y}_{\boldsymbol{q}}(\boldsymbol{q}_2^n,\varphi^n) & -1 & 0 & 0 \\
1 & a_{41}\Delta t \mathrm{Y}_{\boldsymbol{q}}(\boldsymbol{q}_1^n,\varphi^n) & a_{42}\Delta t \mathrm{Y}_{\boldsymbol{q}}(\boldsymbol{q}_2^n,\varphi^n) & a_{43}\Delta t \mathrm{Y}_{\boldsymbol{q}}(\boldsymbol{q}_3^n,\varphi^n) & -1 & 0 \\
1 & b_1\Delta t \mathrm{Y}_{\boldsymbol{q}}(\boldsymbol{q}_1^n,\varphi^n) & b_2\Delta t \mathrm{Y}_{\boldsymbol{q}}(\boldsymbol{q}_2^n,\varphi^n) & b_3\Delta t \mathrm{Y}_{\boldsymbol{q}}(\boldsymbol{q}_3^n,\varphi^n) & b_4\Delta t \mathrm{Y}_{\boldsymbol{q}}(\boldsymbol{q}_4^n,\varphi^n) & -1
\end{bmatrix}, \tag{B6}
$$

$$
\mathrm{L} = \begin{pmatrix}
\delta\boldsymbol{q}^n \\
\delta\boldsymbol{q}_1^n \\
\delta\boldsymbol{q}_2^n \\
\delta\boldsymbol{q}_3^n \\
\delta\boldsymbol{q}_4^n \\
\delta\boldsymbol{q}^{n+1}
\end{pmatrix}
\qquad
\mathrm{M} = -\begin{pmatrix}
0 \\
0 \\
\sum_{j=1}^{1} a_{2j}\Delta t \mathrm{Y}_{\varphi}(\boldsymbol{q}_j^n,\varphi^n)\delta\varphi^n \\
\sum_{j=1}^{2} a_{3j}\Delta t \mathrm{Y}_{\varphi}(\boldsymbol{q}_j^n,\varphi^n)\delta\varphi^n \\
\sum_{j=1}^{3} a_{4j}\Delta t \mathrm{Y}_{\varphi}(\boldsymbol{q}_j^n,\varphi^n)\delta\varphi^n \\
\sum_{i=1}^{4} b_i\Delta t \mathrm{Y}_{\varphi}(\boldsymbol{q}_i^n,\varphi^n)\delta\varphi^n
\end{pmatrix}. \tag{B7}
$$

## B3 Adjoint Equations

Denote the adjoint vector for time step $n$ by $\boldsymbol{N}^n = \left(\boldsymbol{q}^{*^n}\ \boldsymbol{q}_1^{*^n}\ \boldsymbol{q}_2^{*^n}\ \boldsymbol{q}_3^{*^n}\ \boldsymbol{q}_4^{*^n}\ \boldsymbol{q}^{*^{n+1}}\right)$, where $\boldsymbol{q}^{*^n} = [\boldsymbol{\xi}^n, \pi^n, \sigma^n, \eta^n]$ are the adjoint

595 variables associated to the state vector $\boldsymbol{q}^n$. Again in the notation of Yilmaz and Meyers, we define the adjoint variables as the solution of the adjoint equation:

$$
\left(\boldsymbol{N}^n \mathrm{K}^n\right)^T = \boldsymbol{I}_{\boldsymbol{q}}^n = \left(I_{\boldsymbol{q}^n}^n\ \ I_{\boldsymbol{q}_1^n}^n\ \ I_{\boldsymbol{q}_2^n}^n\ \ I_{\boldsymbol{q}_3^n}^n\ \ I_{\boldsymbol{q}_4^n}^n\ \ I_{\boldsymbol{q}^{n+1}}^n\right)^T. \tag{B8}
$$

As opposed to Yilmaz and Meyers, the partial derivatives of the cost function (B4) with respect to the Runge–Kutta stages are now nonzero (due to the more elaborate discretization):

600 $\quad I_{\boldsymbol{q}_i^n}^n = b_i \Delta t J_{\boldsymbol{q}}(\varphi^n, \boldsymbol{q}_i^n) \qquad\qquad i = 1\ldots4,$ $\tag{B9}$

$I_{\boldsymbol{q}^n}^n = I_{\boldsymbol{q}^{n+1}}^n = \boldsymbol{0}.$ $\tag{B10}$

Based on eq. (B8), the temporally discrete adjoint equations become:

$$
\boldsymbol{q}_i^{*^n} = b_i\Delta t\left(\mathrm{Y}_{\boldsymbol{q}}^T(\boldsymbol{q}_i^n,\varphi^n)\boldsymbol{q}^{*^{n+1}} - J_{\boldsymbol{q}}(\boldsymbol{q}_i^n,\varphi^n)\right) + \Delta t\sum_{j=i+1}^{4} a_{ji}\mathrm{Y}_{\boldsymbol{q}}^T(\boldsymbol{q}_i^n,\varphi^n)\boldsymbol{q}_j^{*^n} \qquad i = 1\ldots4, \tag{B11}
$$

$$
\boldsymbol{q}^{*^n} = \boldsymbol{q}^{*^{n+1}} + \sum_{i=1}^{4} \boldsymbol{q}_i^{*^n}. \tag{B12}
$$

605 These equations run backward in time, starting from terminal conditions $\boldsymbol{q}^{*N} = I_{\boldsymbol{q}^N}^N = \boldsymbol{0}$. Since each Runge–Kutta stage contributes to the (discretized) cost function, the cost function gradient $J_{\boldsymbol{q}}$ now appears as a source term in each equation. Note that, structurally, the temporally discrete adjoint equations differ from the forward Runge–Kutta scheme, since the method is not self-adjoint.





Due to the linearity of the adjoint equations, the stage equations from (B12) can be rewritten as follows:

$$\boldsymbol{q}_i^{*^n} = \Delta t \mathrm{Y}_{\boldsymbol{q}}^T(\boldsymbol{q}_i^n, \varphi^n)\Big(b_i \boldsymbol{q}^{*^{n+1}} + \sum_{j=i+1}^4 a_{ji}\boldsymbol{q}_j^{*^n}\Big) - b_i\Delta t J_{\boldsymbol{q}}(\boldsymbol{q}_i^n, \varphi) \qquad\qquad i = 1\dots 4. \tag{B13}$$

To ease the notation in the remainder of the derivation, we introduce the auxiliary variable $\hat{\boldsymbol{q}}_i^{*^n} = [\hat{\boldsymbol{\xi}}_i^n, \hat{\pi}_i^n, \hat{\sigma}_i^n, \hat{\eta}_i^n]$,

$$\hat{\boldsymbol{q}}_i^{*^n} = b_i \boldsymbol{q}^{*^{n+1}} + \sum_{j=i+1}^4 a_{ji}\boldsymbol{q}_j^{*^n} \qquad\qquad i = 1\dots 4. \tag{B14}$$

such that we arrive at the following adjoint equations:

$$\boldsymbol{q}_i^{*^n} = \Delta t \mathrm{Y}_{\boldsymbol{q}}^T(\boldsymbol{q}_i^n, \varphi^n)\hat{\boldsymbol{q}}_i^{*^n} - b_i\Delta t J_{\boldsymbol{q}}(\boldsymbol{q}_i^n, \varphi) \qquad\qquad i = 1\dots 4. \tag{B15}$$

that can be solved backwards, starting from $i = 4$.

For the wind farm power optimization problem at hand, the adjoint Jacobians $\mathrm{Y}_{\boldsymbol{q}}^T$ and $J_{\boldsymbol{q}}$ are exactly equal to the ones used in Munters and Meyers (2018b) where the continuous adjoint approach was used. To apply the temporally discrete adjoint equations to wind farm control problem (1)–(7), we can therefore recycle these expressions, however, here we must also include the extra dependency of the look-up correction factor in (10) on the thrust coefficient and yaw angle. As such, we arrive at the following temporally discrete adjoint Runge–Kutta 4 scheme for the Navier–Stokes equations (where here $V$ and $\overline{V}$ denote the uncorrected and corrected disc-averaged velocities, cf. eq. (10)):

$$\boldsymbol{\xi}_i^n/\Delta t = \Big(-(\nabla \boldsymbol{u}_i^n)^T + (\boldsymbol{u}_i^n \cdot \nabla)\Big)\hat{\boldsymbol{\xi}}_i^n - \nabla\pi_i^n/\rho - \nabla\cdot\boldsymbol{\tau}_{\mathrm{sgs}}^*(\boldsymbol{u}_i^n, \hat{\boldsymbol{\xi}}_i^n) + \sum_{m=1}^{N_t} \boldsymbol{f}_{m,i}^{*^n} \qquad\qquad i = 1\dots 4, \tag{B16}$$

$$\nabla^2\pi_i^n/\rho = \nabla\cdot\Big[\Big(-(\nabla\boldsymbol{u}_i^n)^T + (\boldsymbol{u}_i^n\cdot\nabla)\Big)\hat{\boldsymbol{\xi}}_i^n - \nabla\cdot\boldsymbol{\tau}_{\mathrm{sgs}}^*(\boldsymbol{u}_i^n, \hat{\boldsymbol{\xi}}_i^n) + \sum_{m=1}^{N_t}\boldsymbol{f}_{m,i}^{*^n}\Big] \qquad\qquad i = 1\dots 4, \tag{B17}$$

$$\boldsymbol{\xi}^n = \boldsymbol{\xi}^{n+1} + \sum_{i=1}^4 \boldsymbol{\xi}_i^n, \tag{B18}$$

where (B17) represents the Poisson equation obtained from the time-discrete adjoint momentum equations (B16) for the Runge–Kutta stages. The Poisson equation is solved at every time step and for every stage using a direct method. The adjoint wind farm force in Runge–Kutta stage $i$ in (B16)–(B17) then becomes (with $m$ the turbine number):

$$\boldsymbol{f}_{m,i}^{*^n} = \frac{1}{2}\hat{C}_{T,m}^{'^n}M(\hat{C}_{T,m}^{'^n}, \theta_m^n)\overline{V}_{m,i}^n(3b_i\overline{V}_{m,i}^n - 2\hat{X}_{m,i}^n)\mathcal{R}_m\boldsymbol{e}_{\perp,m}^n \qquad\qquad i = 1\dots 4, \tag{B19}$$

where $\hat{X}_{m,i}^n = \frac{1}{A_m}\int_\Omega \mathcal{R}_m\hat{\boldsymbol{\xi}}_i^n\cdot\boldsymbol{e}_{\perp,m}^n d\boldsymbol{x}$ denotes the disc-averaged adjoint velocity based on $\hat{\boldsymbol{\xi}}_i^n$, and $\overline{V}_{m,i}^n$ the corrected disc-averaged forward velocity based on $\boldsymbol{u}_i^n$ (cf. eq. (10)).

Analogeously, for the thrust coefficient filter and yaw rate equations, we get (i- and j-subscripts denote Runge–Kutta stages, m-subscripts denote turbine numbers):

$$\sigma_{m,i}^n/\Delta t = \frac{1}{\tau}\Big(\hat{\sigma}_{m,i}^n - \frac{1}{2}\overline{V}_{m,i}^{n\,2}(b_i\overline{V}_{m,i}^n - \hat{X}_{m,i}^n)A_m - \frac{1}{2}\hat{C}_{T,m}^{'^n}\overline{V}_{m,i}^n V_{m,i}^n \frac{\partial M(\hat{C}_{T,m}^{'^n}, \theta_m^n)}{\partial\hat{C}_T'}(3b_i\overline{V}_{m,i}^n - 2\hat{X}_{m,i}^n)A_m\Big) \quad i = 1\dots 4, \tag{B20}$$



$$\sigma_m^n = \sigma_m^{n+1} + \sum_{i=1}^{4} \sigma_{m,i}^n, \tag{B21}$$

and

$$\eta_{m,i}^n / \Delta t = -\frac{1}{2} \hat{C}_{T,m}'^n \overline{V}_{m,i}^n \left[ \int_\Omega \left( (3b_i \overline{V}_{m,i}^n - 2\hat{X}_{m,i}^n) M \boldsymbol{u}_i^n - \overline{V}_{m,i}^n \boldsymbol{\xi} \right) \cdot \left( \boldsymbol{e}_{||,m} \mathcal{R}_m + \boldsymbol{e}_{\perp,m} \mathcal{D}_m \right) d\boldsymbol{x} \right]$$

$$- \frac{1}{2} \hat{C}_{T,m}'^n \overline{V}_{m,i}^n V_{m,i}^n \frac{\partial M(\hat{C}_{T,m}'^n, \theta_m^n)}{\partial \theta} (3b_i \overline{V}_{m,i}^n - 2\hat{X}_{m,i}^n) A_m \qquad i = 1\ldots 4, \tag{B22}$$

$$\eta_m^n = \eta_m^{n+1} + \sum_{i=1}^{4} \eta_{m,i}^n. \tag{B23}$$

For a more detailed explanation on all the terms and equations, as well as the derivation of the adjoints, the reader is referred
to Goit and Meyers (2015) and Munters and Meyers (2017, 2018b).

## B4   Adjoint Gradients

The total variation of the cost functional follows from the chain rule:

$$\delta \tilde{\mathcal{J}}^N = \sum_{n=1}^{N} \delta I^n = \sum_{n=1}^{N} \left( \mathrm{L}^n \right)^T \boldsymbol{I}_{\boldsymbol{q}}^n + I_\varphi \delta \varphi^n. \tag{B24}$$

We can now plug in the adjoint equation (B8) and then the linarization (B5), such that

$$\delta \tilde{\mathcal{J}}^N = \sum_{n=1}^{N} \delta I^n = \sum_{n=1}^{N} \left( \mathrm{L}^n \right)^T \left( \mathrm{K}^n \right)^T \left( \boldsymbol{N}^n \right)^T + I_\varphi \delta \varphi^n = \sum_{n=1}^{N} \left( \mathrm{M}^n \right)^T \left( \boldsymbol{N}^n \right)^T + I_\varphi \delta \varphi^n. \tag{B25}$$

The derived expression hence does not require the forward sensitivity matrix L, which would otherwise have to be determined
for every control perturbation making the computation very expensive. From (B25), we can derive the gradient in any control
direction. In practice, the L-BFGS-B library needs the gradient for every control variable $\delta \varphi^n$, which amounts to:

$$\frac{\partial \tilde{\mathcal{J}}^N}{\partial \varphi^n} = -\Delta t \sum_{i=1}^{4} \left( \sum_{j=1}^{i-1} \left( a_{ij} \mathrm{Y}_\varphi^T (\boldsymbol{q}_j^n, \varphi^n) \boldsymbol{q}_i^{*n} \right) + b_i \left( \mathrm{Y}_\varphi^T (\boldsymbol{q}_i^n, \varphi^n) \boldsymbol{q}^{*n+1} - J_\varphi (\boldsymbol{q}_i^n, \varphi^n) \right) \right). \tag{B26}$$

Note that (B26) is the exact gradient of the discretized objective function in (B2), which converges to the gradient of the
continuous problem in the limit of $\Delta t \to 0$ (Giles and Pierce, 2000).

Applied to the wind farm control problem (1)–(7), we arrive at the following simplified expressions for the gradient of the
wind farm power objective function with respect to the thrust coefficients and yaw rates (i-subscripts denote Runge–Kutta
stages, m-subscripts denote turbine numbers):

$$\nabla_{\varphi_m} \tilde{\mathcal{J}}^N = \begin{pmatrix} \frac{\partial \tilde{\mathcal{J}}^N}{\partial C_{T,m}'^n} \\ \frac{\partial \tilde{\mathcal{J}}^N}{\partial w_m^n} \end{pmatrix} = \begin{pmatrix} -\Delta t \sum_{i=1}^{4} \left( b_i \sigma_m^{n+1} + \sum_{j=1}^{i-1} a_{ij} \sigma_{m,i}^n \right) \\ -\Delta t \sum_{i=1}^{4} \left( b_i \eta_m^{n+1} + \sum_{j=1}^{i-1} a_{ij} \eta_{m,i}^n \right) \end{pmatrix}. \tag{B27}$$



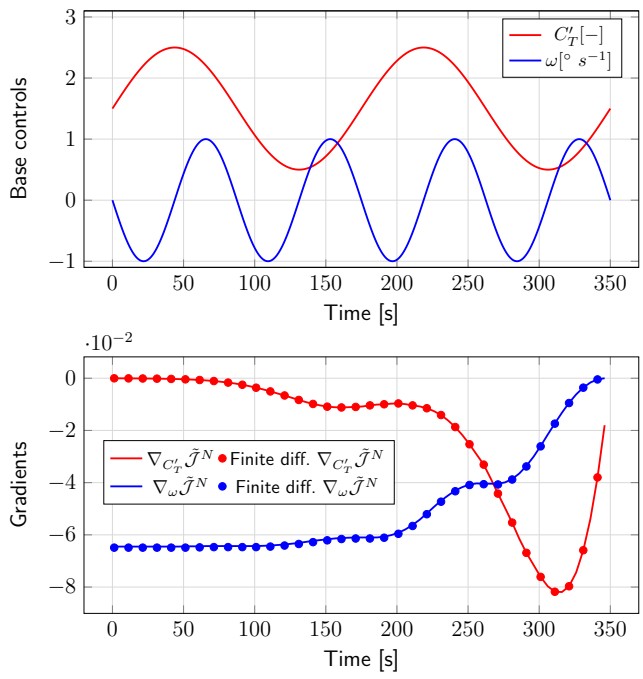

**Figure B1.** Baseline controls for which the gradient is computed and adjoint gradients versus finite difference verification.

### B5 Gradient Verification

In this section, expression (B27) for the gradient of the objective function, resulting from the temporally discrete adjoint method, is validated via finite differences. For the gradient verification, we consider the same numerical setup from Sect. 3.1. Via finite differences, the Gateaux derivative of the objective function in the direction $\delta\varphi$ is approximated as:

$$\left(\nabla\tilde{\mathcal{J}}^N, \delta\varphi\right) \approx \frac{\tilde{\mathcal{J}}^N(\varphi + \alpha\delta\varphi) - \tilde{\mathcal{J}}^N(\varphi)}{\alpha}, \tag{B28}$$

where we set $\alpha = 10^{-6}$. To limit computational costs, we only examine the gradients for turbines R1C1 and R8C4 (respectively front and last row in Fig. 6) for a control horizon $T = 350$ s (the longest horizon considered in this work) and for a limited amount of time instants. The baseline controls, resulting adjoint gradients and finite difference verification are shown in Fig. B1.

### Appendix C: Yaw and Induction Characteristics

Figure C1 and C2 and Fig. C3 and C4 display the time evolution of the optimized filtered thrust coefficients $\hat{C}'_T$ and yaw angles $\theta$ for the eight turbines in column C1 for respectively grid level 1 and grid level 2. As in Sect. 4.4 for grid level 0, only the turbines in column C1 are shown for cases 1, 4, 5, 7, 8, 9 and 10. Regarding the influence of the receding-horizon parameters, the conclusions are similar as for grid level 0. Note that for level 1 and level 2, the dips in $\hat{C}'_T$ are more prominent than for grid level 0 in Sect. 4.4. However, in each of the cases, yaw control remains the dominant control mechanism.



**Figure C1.** Time evolution of filtered thrust coefficients $\hat{C}'_T$ for turbine column C1 for different optimal control cases for grid level 1.



**Figure C2.** Time evolution of yaw angles $\theta$ for turbine column C1 for different optimal control cases for grid level 1.



**Figure C3.** Time evolution of filtered thrust coefficients $\hat{C}'_T$ for turbine column C1 for different optimal control cases for grid level 2.



**Figure C4.** Time evolution of yaw angles $\theta$ for turbine column C1 for different optimal control cases for grid level 2.



*Data availability.* Datasets supporting this article will be made available in the KU Leuven Research Data Repository (KU Leuven RDR). The datasets include optimized control and power histories from the control model and wind farm emulator for the cases reported in this paper.

*Author contributions.* NJ and JM jointly set up the simulation studies in the current work. NJ performed code implementations and carried out the simulations. NJ and JM jointly wrote the manuscript.

*Competing interests.* At least one of the (co-)authors is a member of the editorial board of Wind Energy Science. The peer-review process was guided by an independent editor, and the authors have no other competing interests to declare.

*Acknowledgements.* The authors acknowledge support from the Research Foundation - Flanders (FWO), grant number 1SD4121N. The computational resources and services used in this work were provided by the VSC (Flemish Supercomputer Center), funded by the Research Foundation - Flanders and the Flemish Government, department EWI. NJ used ChatGPT to reformulate some phrases in the abstract,
introduction and discussion and conclusion sections starting from an own earlier formulation.



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
