# Peer review of "Towards real-time optimal control of wind farms using large-eddy simulations"

_Wind Energy Science, 2023_

## Author Comment (AC1)

**Towards real-time optimal control of wind farms using large-eddy simulations**

Anonymous Referee #1 comments

We thank the reviewer for the careful assessment of our work. We readily addressed the issues raised by the reviewer as further discussed below. Answers to reviewer comments are indicated in red, textual changes in the revised manuscript are indicated in blue between double quotes "".

In this study, the author investigates real-time optimal control of wind farms using large-eddy simulations. Overall, this manuscript is of good quality and novelty. However, I still have some doubts that require further clarification in the current manuscript by the authors.

1. In section 2.3, the authors discussed the turbine modelling used in this study and proposed a look-up table approach as an ad-hoc solution to fix the low-resolution issue ( 2 grid points across the rotor diameters). However, the main concern of this reviewer is that the look-up table in Appendix A is constructed based on **uniform inflow**, neglecting the effects of turbulent inflow and vertical wind shear that are critical to the power of wind turbines in a wind farm). As a quick check, this reviewer would like to ask the authors to examine the correcting factor M in a turbulent inflow generated by the precursor simulation and check if it significantly differs from the uniform inflow results.

   The look-up table is indeed constructed based on uniform inflow and included in the proposed control model that is used throughout the optimization (and prediction). All control signals are validated in fine-grid LES (in the wind farm emulator), with turbulent inflow. As such, the significant power gains that were reported in the paper already provide a proof-of-concept for the look-up table approach.

   Also, we computed the correction factor so that the steady state power on the coarse grids (after correction) matches that of the fine-grid reference for uniform inflow. For turbulent inflow, one could devise a similar strategy by matching the average power (since there is no "steady state" power) over some time horizon. However, additional input parameters, such as shear and turbulence intensity, may/should then also be considered. Moreover, the computation of the look-up table comprises an iterative procedure with LES simulations (that has to be repeated for every entry, so for every combination of $\hat{C}'_T$ and $\theta$), which would become quite cumbersome for turbulent inflow.

   We can, however, quantify the performance of the look-up table approach by comparing the corrected and uncorrected power on the coarse grids to that of the fine-grid emulator for turbulent inflow. In the revised manuscript, in App. A, we included such an analysis and show that the correction factor reduces the prediction error on the mean power production: "Finally, we also briefly examine the performance of the look-up table approach for turbulent inflow. To that end, as an example, we take the setup of case 1 from Tab. 2 and Tab. 3 on grid level 0. For $\hat{C}'_T = 2.0$ and $\theta = 0°$, Fig. A1 shows the corrected (i.e. using the look-up table approach) and uncorrected power predictions on the coarse grids compared to the fine-grid reference for a horizon of 300 s. With the look-up table approach, the error on the average power prediction is decreased by 7%, i.e. from an overestimation of 21% for the uncorrected prediction to an underestimation of 14% for the corrected one."

2. In section 4.3.1, the authors discussed the computational cost. However, since the simulation in this study required a precursor simulation to drive the flow, this reviewer would like to ask the authors to discuss the cost of precursor simulation in this analysis.

   The computational cost of the precursor simulation is not reported in the manuscript, because the precursor is only required to generate turbulent inflow for the fine-grid emulator. In the propagator and controller, we select an upstream domain length that is large enough to ensure that the inflow never reaches the front-row turbines, meaning that the precursor is redundant there (cf. Sect. 2.5.1). In Sect. 4.3.1, we only consider the computational speed of the optimization (which has to run in real-time); the precursor simulation does not need to

run in real-time, so its computational cost is irrelevant in the analysis of the real-time factor. In the revised manuscript on p.19, we clearified this as follows:

"Remark that we only consider the computational cost of the real-time computations (the fine-grid emulator, precursor, ... do not need to run in real-time and are hence not included in the analysis of the real-time factor)".

As described in the manuscript, the precursor data was taken from Munters e.a. (2019) [a], the reader is referred there for details on the precursor simulation.

[a] Munters, Wim, Sood, Ishaan, & Meyers, Johan. (2019). Precursor dataset PDk [Data set]. Zenodo. https://doi.org/10.5281/zenodo.2650100"

**Towards real-time optimal control of wind farms using large-eddy simulations**

Anonymous Referee #2 comments

We thank the reviewer for the careful assessment of our work. We readily addressed the issues raised by the reviewer as further discussed below. Answers to reviewer comments are indicated in red, textual changes in the revised manuscript are indicated in blue between double quotes "".

The paper discusses the implementation and testing of wind farm control strategies (combination of wake steering by yawing and reduction of axial induction) synthesized by means of an LES-based control model. Specifically, the authors aim to investigate the impact of the receding horizon parameters, control update frequency, and grid resolution on the computational time. The final objective of their efforts is to enable a real-time implementation of the proposed control methodology. All results are obtained in simulations, with a fine-grid LES model used as plant emulator.

The paper is interesting, describes a novel approach, and is well-written. I have a single major comment.

The gains of case 4 (circa 50%) are much higher than the gains with steady yaw control (circa 10%, as also reported in Sood and Meyers, 2022). The authors claim that "Simulations (not shown here) suggest that yaw control only (i.e. disabling induction control) does not entail a significant performance reduction.", which implies that the large difference between the gains of case 4 and of Sood and Meyers is not due to enabling the reduction of induction control in case 4. The authors also claim that case 4 is characterized by quasi-static yawing behavior. For this case, the yaw angles, indeed, do not exhibit major variations, but rather minorly oscillate around the quasi-static optimal yaw angles. It is reasonable to expect the quasi-static optimal yaw angles to be close to the optimal yaw angles of the steady yaw control of Sood and Meyers (in this regard, it would be helpful to include the optimal yaw angles of Sood and Meyers in Figure 13). , I, therefore, wonder what the reason for the very big differences between the achieved gains of case 4 and of the steady yaw control. Please argue on this.

The gain of 50 % over the steady yaw controller from Sood and Meyers (2022) effectively originates from the dynamic yaw steering. In the revised manuscript, we included the steady yaw angles in App. D. In their framework, most turbines remain unyawed or only exhibit minor yaw angles of $-1$ or $-2°$ degrees, except for the front-row turbines that are yawed to $\pm 30°$. As suggested by the reviewer, in our simulations, the quasi-static yawing behavior for the front-row turbines indeed matches the $\pm 30°$ yaw angles from Sood and Meyers. However, as shown in Fig. 13 for case 4 and case 8, we also observe a reversal of the quasi-static yaw angles from $30°$ to $-30°$ and vice versa to optimally react to the turbulent inflow.
The major difference compared to Sood and Meyers resides in the (dynamic) yawing of the downstream turbines. As shown in Figure 13 for case 4 in particular, all downstream turbines (from R2–R8) are effectively yawed, whereas this was not the case in Sood and Meyers.
On the one hand, the quasi-static yawing of the front-row turbines persists downstream in rows R2–R4/R5 (although the amplitude of the oscillations increases in response to the increasing local unsteadiness of the flow). Consequently, also the wakes from those turbines are steered away from the downstream turbines in an "optimal" way (i.e. tailored to the turbulent inflow), entailing significant power gains that are not captured by the framework of Sood and Meyers.
On the other hand, towards the end of the farm, the flow is very unsteady due to the superimposed wakes from the upstream turbines, rendering quasi-static yawing rather disadventageous. However, these turbines are still yawed dynamically, mostly aligning them to the incoming flow. The mean angles of approx. $0°$ are similar to those of Sood and Meyers, but the oscillations in response to the turbulent inflow (that again entail significant power gains) are not possible with a static yaw controller.

Finally, it must also be noted that the proposed LES control model is able to account for secondary steering effects, whereas these are not included the Bastankhah wake model that was used in Sood and Meyers to determine the yaw setpoints.

In conclusion, the dynamic yawing in response to the unsteady inflow, not only for front-row turbines but throughout the whole farm, explain the significant power gains of our LES-based controller compared to the steady yaw controller from Sood and Meyers.

In the revised paper on p. 24, we included the following discussion:

"The substantial power gains compared to the steady yaw controller from Sood and Meyers (2022) originate from the dynamic yaw steering throughout the whole farm in response to the turbulent inflow. As shown in App. D for the framework of Sood and Meyers, only front-row turbines are effectively yawed to $\pm 30°$, whereas we also observe significant yawing in the downstream regions of the farm. On the one hand, for case 4 in particular, there is the quasi-static wake steering — with occasional turnovers from $30°$ to $-30°$ and vice versa depending on the inflow — that persists downstream in turbine rows R2–R4/R5. On the other hand, towards the end of the farm, we also observe dynamic yawing around a mean angle of approx. $0°$ to optimally align the turbines to the turbulent inflow that has become increasingly unsteady due to the superimposed wakes from the upstream turbines. This also entails a significant power gain that cannot be captured with a steady yaw controller. Finally, we note that the proposed LES-based controller is able to account for secondary steering effects that are not included in the Bastankhah wake model that was used to optimize the yaw setpoints in Sood and Meyers."

Minor comments:

- Figure 9, error in caption: PDE, not L-BFGS-B
  In the revised manuscript, we clarified the caption as follows:
  "Cost function vs. number of PDE evaluations for cases 4, 7, 9 and 10 (with $T_A = 50$ s) on each grid level for the optimization window starting at $t+T_A = 750$ s. Circles mark L-BFGS-B iterations. For every grid level $i$, the cost function $\hat{\mathcal{J}}^i$ is scaled by $\hat{\mathcal{J}}_0^i$ of the first iteration."

- Line 516, typo in "investigated"
  We corrected this in the revised manuscript.